# CONFIDENT SINKHORN ALLOCATION FOR PSEUDO-LABELING

## ABSTRACT

Semi-supervised learning is a critical tool in reducing machine learning's dependence on labeled data. It has been successfully applied to structured data, such as images and natural language, by exploiting the inherent spatial and semantic structure therein with pretrained models or data augmentation. These methods are not applicable, however, when the data does not have the appropriate structure, or invariances. Due to their simplicity, pseudo-labeling (PL) methods can be widely used without any domain assumptions. However, PL is sensitive to a threshold and can perform poorly if wrong assignments are made due to overconfidence. This paper studies theoretically the role of uncertainty to pseudo-labeling and proposes Confident Sinkhorn Allocation (CSA), which identifies the best pseudo-label allocation via optimal transport to only samples with high confidence scores. CSA outperforms the current state-of-the-art in this practically important area of semi-supervised learning. Additionally, we propose to use the Integral Probability Metrics to extend and improve the existing PAC-Bayes bound which relies on the Kullback-Leibler (KL) divergence, for ensemble models.

## 1 INTRODUCTION

The impact of machine learning continues to grow in fields as disparate as biology (Libbrecht and Noble, 2015; Tunyasuvunakool et al., 2021), quantum technology (Biamonte et al., 2017; van Esbroeck et al., 2020; Nguyen et al., 2021), brain stimulation (Boutet et al., 2021; van Bueren et al., 2021), and computer vision (Esteva et al., 2021; Yoon et al., 2022; Long et al., 2022). Much of this impact depends on the availability of large numbers of annotated examples for the machine learning models to be trained on. The data annotation task by which such labeled data is created is often expensive, and sometimes impossible, however. Rare genetic diseases, stock market events, and cyber-security threats, for example, are hard to annotate due to the volumes of data involved, the rate at which the significant characteristics change, or both.

**Related work.** Fortunately, for some classification tasks, we can overcome a scarcity of labeled data using semi-supervised learning (SSL) (Zhu, 2005; Huang et al., 2021; Killamsetty et al., 2021; Olsson et al., 2021). SSL exploits an additional set of unlabeled data with the goal of improving on the performance that might be achieved using labeled data alone (Lee et al., 2019; Carmon et al., 2019; Ren et al., 2020; Islam et al., 2021).

Domain specific: Semi-supervised learning for image and language data has made rapid progress (Oymak and Gulcu, 2021; Zhou, 2021; Sohn et al., 2020) largely by exploiting the inherent spatial and semantic structure of images (Komodakis and Gidaris, 2018) and language (Kenton and Toutanova, 2019). This is achieved typically either using pretext tasks (Komodakis and Gidaris, 2018; Alexey et al., 2016) or contrastive learning (Van den Oord et al., 2018; Chen et al., 2020). Both approaches assume that specific transformations applied to each data element will not affect the associated label.

Greedy pseudo-labeling: A simple and effective SSL method is pseudo-labeling (PL) (Lee et al., 2013), which generates 'pseudo-labels' for unlabeled samples using a model trained on labeled data. A label $k$ is assigned to an unlabeled sample $\mathbf{x}_i$ if the predicted class probability is over a predefined threshold $\gamma$ as

$$y_i^k = \mathbb{1}\Big[p(y_i = k \mid \mathbf{x}_i) \geq \gamma\Big] \tag{1}$$

Table 1: Comparison with the related approaches in terms of properties and their relative trade-offs.

| Algorithms | Not domain specific | Uncertainty consideration | Non-greedy |
|---|:---:|:---:|:---:|
| Pseudo-Labeling (Lee et al., 2013) | ✓ | ✗ | ✗ |
| FlexMatch (Zhang et al., 2021) | ✓ | ✗ | ✗ |
| Vime (Yoon et al., 2020) | ✗ | ✗ | NA |
| MixMatch (Berthelot et al., 2019) | ✗ | ✗ | NA |
| FixMatch (Sohn et al., 2020) | ✗ | ✗ | NA |
| UPS (Rizve et al., 2021) | ✓ | ✓ | ✗ |
| SLA (Tai et al., 2021) | ✓ | ✗ | ✓ |
| **CSA** | ✓ | ✓ | ✓ |

where $\gamma \in [0, 1]$ is a threshold used to produce hard labels and $p(y_i = k \mid \mathbf{x}_i)$ is the predictive probability of the $i$-th data point belonging to the class $k$. A classifier can then be trained using both the original labeled data and the newly pseudo-labeled data. Pseudo-labeling is naturally an iterative process, with the next round of pseudo-labels being generated using the most-recently trained classifier. The key advantage of pseudo-labeling is that it does not inherently require domain assumptions and can be applied to most domains, including tabular data where domain knowledge is not available, as opposed to images and languages.

Greedy PL with uncertainty: Rizve et al. (2021) propose an uncertainty-aware pseudo-label selection (UPS) that aims to reduce the noise in the training process by using the uncertainty score – together with the probability score for making assignments:

$$y_i^k = \mathbb{1}\Big[p(y_i = k \mid \mathbf{x}_i) \geq \gamma\Big] \, \mathbb{1}\Big[\mathcal{U}\big(p(y_i = k \mid \mathbf{x}_i) \leq \gamma_u\big)\Big] \tag{2}$$

where $\gamma_u$ is an additional threshold on the uncertainty level and $\mathcal{U}(p)$ is the uncertainty of a prediction $p$. As shown in Rizve et al. (2021), selecting predictions with low uncertainties greatly reduces the effect of poor calibration, thus improving robustness and generalization.

However, the aforementioned works in PL are *greedy* in assigning the labels by simply comparing the prediction value against a predefined threshold $\gamma$ irrespective of the relative prediction values across samples and classes. Such greedy strategies will be sensitive to the choice of thresholds, and once a sample has been misclassified the error can cascade.

Non-greedy pseudo-labeling: FlexMatch (Zhang et al., 2021) adaptively selects a threshold $\gamma_k$ for each class based on the level of difficulty. This threshold is adapted using the predictions across classes. However, the selection process is still heuristic in comparing the prediction score with an adjusted threshold. Recently, Tai et al. (2021) provide a novel view in connecting the pseudo-labeling assignment task to optimal transport problem, called SLA, which inspires our work. SLA and FlexMatch are better than existing PL in that their *non-greedy* label assignments not only use the single prediction value but also consider the relative importance of this value across rows and columns in a holistic way. However, both SLA and FlexMatch can overconfidently assign labels to noise samples and have not considered utilizing uncertainty values in making assignments.

**Contributions.** We propose here a semi-supervised learning method that does not require any domain-specific assumption for the data. We hypothesize that this is by far the most common case for the vast volumes of data that exist. Our method Confident Sinkhorn Allocation (CSA) is theoretically driven by the role of uncertainty in robust label assignment in SSL. CSA utilizes Sinkhorn's algorithm (Cuturi, 2013) to assign labels to only the data samples with high confidence scores. By learning the label assignment with optimal transport, CSA eliminates the need to predefine the heuristic thresholds used in existing pseudo-labeling methods, which can be greedy. The proposed CSA is widely applicable to various data domains, and could be used in concert with consistency-based approaches (Sohn et al., 2020), but is particularly useful for data domain where pretext tasks and data augmentation are not applicable, such as tabular data. Finally, we study theoretically the pseudo-labelling process when training on labeled set and predicting unlabeled data using a PAC-Bayes generalization bound. Particularly, we extend and improve the existing PAC-Bayes bounds with Kullback-Leibler (KL) divergence by presenting the first result making use of the Integral Probability Metrics (see Sec. 2.4).

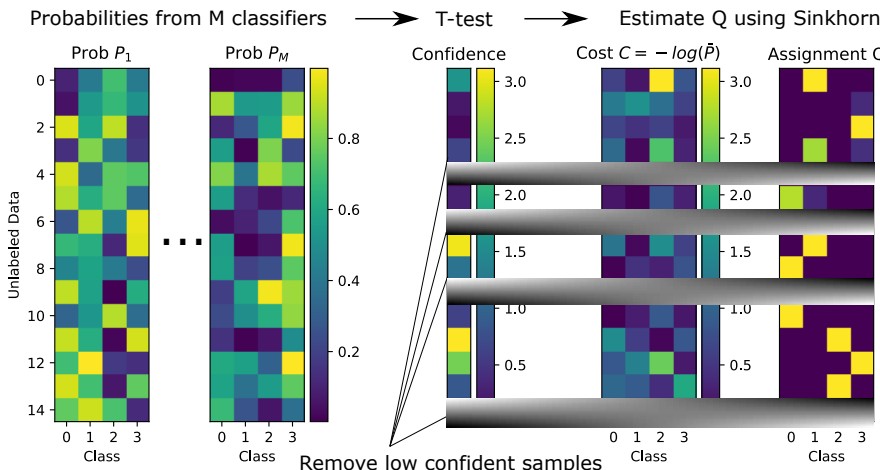

Figure 1: A depiction of CSA in application. We estimate the ensemble of predictions $P$ on unlabeled data using $M$ classifiers which can result in different probabilities. We then identify high-confidence samples by performing a T-test. Next, we estimate the label assignment $Q$ using Sinkhorn's algorithm. The cost $C$ of the optimal transport problem is the negative of the probability averaging across classifiers, $C = -\log \bar{P}$. We repeat the process on the remaining unlabeled data as required.

## 2 CONFIDENT SINKHORN ALLOCATION (CSA)

We consider the semi-supervised learning setting whereby we have access to a dataset consisting of labeled examples $\mathcal{D}_l = \{\mathbf{x}_i, y_i\}_{i=1}^{N_l}$, and unlabeled examples $\mathcal{D}_u = \{\mathbf{x}_i\}_{i=1}^{N_u}$ where $\mathbf{x}_i \in \mathbb{R}^d$ and $y_i \in \mathcal{Y} = \{1, \ldots, K\}$. We define also $\mathcal{X} = \{\mathbf{x}_i\}, i = \{1, \ldots, N_l + N_u\}$. Our goal is to utilize $\mathcal{D}_l \cup \mathcal{D}_u$ to learn a predictor $f : \mathcal{X} \to \mathcal{Y}$ that is more accurate than a predictor trained using labeled data $\mathcal{D}_l$ alone. The summary of our notations is provided in Appendix Table 4.

Generating high-quality pseudo-labels is critical to the classification performance, as erroneous label assignment can quickly lead the iterative pseudo-labeling process astray. We provide in Sec. 2.1 a theoretical analysis of the role and impact of uncertainty in pseudo-labeling, the first such analysis as far as we are aware. Based on the theoretical result, we propose two approaches to identify the high-confidence samples for assigning labels and use the Sinkhorn's algorithm to find the best label assignment. We name the proposed algorithm Confident Sinkhorn Allocation (CSA). We provide a diagram demonstrating CSA in Fig. 1, and a comparison against related algorithms in Table 1.

### 2.1 ANALYSIS OF THE EFFECT OF UNCERTAINTY IN PL

Our theoretical result extends the result from Theorem 1 of Yang and Xu (2020) in three folds: (i) generalizing from binary to multi classification; (ii) extending from one dimensional to multi-dimensional input; and (iii) taking into account the uncertainty for pseudo-labeling.

We consider a multi-classification problem with the generating feature $\mathbf{x} \mid y = k \overset{\text{iid}}{\sim} \mathcal{N}(\mu_k, \Lambda)$ where $\mu_k \in \mathbb{R}^d$ is the means of the $k$-th classes, $\Lambda$ is the covariance matrix defining the variation of the data noisiness across input dimensions. For simplicity in the analysis, we define $\Lambda = \text{diag}([\sigma_1^2, \ldots \sigma_d^2])$ as a $\mathbb{R}^{d \times d}$ diagonal matrix which does not change with different classes $k$. For the analysis, we make a popular assumption that input features $\mathbf{x}_i \in \mathcal{D}_l$ and $\mathcal{D}_u$ are sampled i.i.d. from a feature distribution $P_X$, and the labeled data pairs $(\mathbf{x}_i, y_i)$ in $\mathcal{D}_l$ are drawn from a joint distribution $P_{X,Y}$.

Let $\{\tilde{X}_i^k\}_{i=1}^{\tilde{n}_k}$ be the set of unlabeled data whose pseudo-label is $k$, respectively. Let $\{I_i^k\}_{i=1}^{\tilde{n}_k}$ be the binary indicator of correct assignments, such as if $I_i^k = 1$, then $\tilde{X}_i^k \sim \mathcal{N}(\mu_k, \Lambda)$.

We assume the binary indicators $I_i^k$ to be generated from a pseudo-labeling classifier with the corresponding expectations $\mathbb{E}(I^k)$ and variances of $\text{Var}(I^k)$ for the generating processes.

In uncertainty estimation literature (Der Kiureghian and Ditlevsen, 2009; Hüllermeier and Waegeman, 2019), $\Lambda$ is called the *aleatoric uncertainty* (from the observations) and $\text{Var}(I^k)$ are the *epistemic uncertainty* (from the model knowledge).

Instead of using linear classifier in the binary case as in Yang and Xu (2020), here we consider a set of probabilistic classifiers: $f_k(\mathbf{x}_i) := \mathcal{N}(\mathbf{x}_i|\hat{\theta}_k, \Lambda), \forall k = \{1, ..., K\}$ to classify $y_i = k$ such that $k = \arg\max_{j \in [1,...,K]} f_j(\mathbf{x}_i)$. Given the above setup, it is natural to construct our estimate as $\hat{\theta}_k = \frac{\sum_{i=1}^{\tilde{n}_k} \tilde{X}_i^k}{\tilde{n}_k}$ via the extra unlabeled data. We aim to characterize the estimation error $\sum_{k=1}^{K} |\hat{\theta}_k - \mu_k|$.

**Theorem 1.** *For $\delta > 0$, our estimate satisfies $\sum_{k=1}^{K} |\hat{\theta}_k - \mu_k| \leq \delta$ with a probability at least $1 - 2\sum_{k=1}^{K}\sum_{j=1}^{d} \exp\left\{ -\frac{\delta^2 \tilde{n}_k}{8\sigma_j^2} \right\} - \sum_{k=1}^{K} \frac{4Var(I^k)}{\delta^2} |\mu_k - \mu_{\backslash k}|$ where we denote $\mu_{\backslash k} = \mu_j \mid \exists j \in \{1, ..., K\} \backslash k$.*

*Proof.* See Appendix A.1. □

**Interpretation.** The theoretical result reveals several interesting aspects. First, we show that more number of unlabeled data $\tilde{n}_k \uparrow$ is useful for a good estimation. We empirically validate this property in Fig. 5. Second, less number of classes $K \downarrow$ and/or less number of input dimensions $d \downarrow$ will make the estimation easier. Finally, our analysis takes a step further to show that both aleatoric uncertainty $\sigma_j^2 \uparrow$ and epistemic uncertainty $\text{Var}(I^k)$ can reduce the probability of obtaining a good estimation. In other words, less uncertainty is more helpful. To the best of our knowledge, we are *the first* to theoretically characterize the above properties in pseudo-labeling. The theoretical result in Yang and Xu (2020) is only applicable for binary classification and the role of uncertainty has not been studied.

## 2.2 IDENTIFYING HIGH-CONFIDENCE SAMPLES FOR LABELING

Machine learning predictions vary with different choices of hyperparameters. Under these variations, it is unclear to identify the most probable class to assign labels for some data points. Not only the predictive values but also the ranking order of these predictive samples vary. The variations in the prediction can be explained due to (i) the uncertainties of the prediction coming from the noise observations (aleatoric uncertainty) and model knowledge (epistemic uncertainty) (Hüllermeier and Waegeman, 2019) and (ii) the gap between the highest and second-highest scores is small. These bring a challenging fact that the highest score class can be changed with a different set of hyperparameters. This leads to the confusion for assigning pseudo-labels because (i) the best set of hyperparameter is unknown given limited labeled data and (ii) we consider ensemble learning setting wherein multiple models are used together.

To address the two challenges above, we are motivated by the theoretical result in Theorem 1 that the data sample noise and uncertainty can worsen the estimated classifier. Therefore, we select to assign labels to only samples in which the most probable class is statistically significant than the second-most probable class. [1]

For this purpose, we propose to measure the confidence level by performing Welch's T-test which is the recommended approach when the variances are different between the groups (Ruxton, 2006; Delacre et al., 2017).

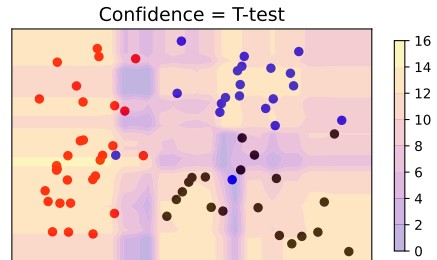

Confidence = T-test

Figure 2: Examples of T-test for estimating the confidence level on $K = 3$ classes. The yellow area indicates high confidence. We exclude the samples falling into the dark area where the value of T-test less than 2.

**Welch's T-test.** For each data point $i$, we define two empirical distributions (see Appendix A.5) of predicting the highest $\mathcal{N}(\mu_{i,\diamond}, \sigma_{i,\diamond}^2)$ and second-highest class $\mathcal{N}(\mu_{i,\oslash}, \sigma_{i,\oslash}^2)$,[2] estimated from the predictive probability across $M$ classifiers, such as $\mu_{i,\diamond} = \frac{1}{M}\sum_{m=1}^{M} p_m(y = \diamond \mid \mathbf{x}_i)$ and $\mu_{i,\oslash} = \frac{1}{M}\sum_{m=1}^{M} p_m(y = \oslash \mid \mathbf{x}_i)$ are the empirical means; $\sigma_{i,\diamond}^2$ and $\sigma_{i,\oslash}^2$ are the variances. We consider Welch's T-test (Welch, 1947) to compare the statistical significance of two empirical distributions:

---

[1] (Rizve et al., 2021) have considered related idea of using uncertainty into PL without theoretical justification.

[2] Denote the highest score class by $\diamond := \diamond(i)$ and second-highest score class by $\oslash := \oslash(i)$ for a data point $i$.

$$\text{T-value}(\mathbf{x}_i) = \frac{\mu_{i,\diamond} - \mu_{i,\oslash}}{\sqrt{\left(\sigma_{i,\diamond}^2 + \sigma_{i,\oslash}^2\right)/M}}. \tag{3}$$

The degree of freedom for the above statistical significance is $(M-1)(\sigma_1^2 + \sigma_2^2)^2/(\sigma_1^4 + \sigma_2^4)$. As a standard practice in statistical testing (Neyman and Pearson, 1933), if p-value is less than or equal to the significance level of 0.05, the two distributions are different. By calculating the degree of freedom and looking at the T-value distribution, this significant level of 0.05 is equivalent to a threshold of 2 for the T-value. Therefore, we get the following rules if the T-value is less than 2, the two considered classes are from the same distribution – *eg.*, the sample might fall into the dark area in *right* Fig. 2. Thus, we exclude a sample from assigning labels when its T-value is less than 2.The estimation of the T-test above encourages separation between the highest and second-highest classes relates to entropy minimization (Grandvalet and Bengio, 2004) which encourages a classifier to output low entropy predictions on unlabeled data. We visualize the confidence estimated by T-test in Fig. 2.

## 2.3 Optimal Transport Assignment

We use the confidence scores to filter out less confident data points and assign labels to the remaining. Particularly, we follow the novel view in Tai et al. (2021) to interpret the label assignment process as an optimal transportation problem (Villani et al., 2009; Peyré et al., 2019; El Hamri et al., 2021) between examples and classes, wherein the cost of assigning an example to a class is dictated by the predictions of the classifier.

Let us denote $N \leq N_u$ be the number of accepted points, *eg.* by T-test, from the unlabeled set. Let define an assignment matrix $Q \in \mathbb{R}^{N \times K}$ of $N$ unlabeled data points to $K$ classes such that we assign $\mathbf{x}_i$ to a class $k$ if $Q_{i,k} > 0$. We learn an assignment $Q$ that minimizes the total assignment cost $\sum_{ik} Q_{ik} C_{ik}$ where $C_{ik}$ is the cost of assigning an example $i$ to a class $k$ given by the corresponding negative probability as used in Tai et al. (2021), i.e., $C_{ik} := -\log p(y_i = k \mid \mathbf{x}_i)$ where $0 \leq p(y_i = k \mid \mathbf{x}_i) \leq 1$

$$\begin{aligned}
\text{minimize}_Q \quad & \langle Q, C \rangle & (4) \\
\text{s.t.} \quad & Q_{ik} \geq 0 & (5) \\
& Q\mathbf{1}_K \leq \mathbf{1}_N & (6) \\
& Q^T \mathbf{1}_N \leq N\mathbf{w}_+ & (7) \\
& \mathbf{1}_N^T Q \mathbf{1}_K \geq N\rho \mathbf{w}_-^T \mathbf{1}_K & (8)
\end{aligned}$$

$$\begin{aligned}
\text{minimize}_{Q,\mathbf{u},\mathbf{v},\tau} \quad & \langle Q, C \rangle & (9) \\
\text{s.t.} \quad & Q_{ik} \geq 0, \mathbf{u} \succeq 0, \mathbf{v} \succeq 0, \tau \geq 0 & (10) \\
& Q\mathbf{1}_K + \mathbf{u} = \mathbf{1}_N & (11) \\
& Q^T \mathbf{1}_N + \mathbf{v} = \mathbf{w}_+ N & (12) \\
& \mathbf{u}^T \mathbf{1}_N + \tau = N(1 - \rho \mathbf{w}_-^T \mathbf{1}_K) & (13) \\
& \mathbf{v}^T \mathbf{1}_K + \tau = N(\mathbf{w}_+^T \mathbf{1}_K - \rho \mathbf{w}_-^T \mathbf{1}_K) & (14)
\end{aligned}$$

where $\mathbf{1}_K$ and $\mathbf{1}_N$ are the vectors one in $K$ and $N$ dimensions, respectively; $\rho \in [0,1]$ is the fraction of assigned label, i.e., $\rho = 1$ is full allocation; $\mathbf{w}_+, \mathbf{w}_- \in \mathbb{R}^k$ are the vectors of upper and lower bound assignment per class which can be estimated empirically from the class label frequency in the training data or from prior knowledge.

Our formulation has been motivated and slightly modified from the original SLA (Tai et al., 2021) to introduce the lower bound $\mathbf{w}_-$ specifying the minimum number of data points to be assigned in each class. We refer to Appendix B.6 for the ablation study on the effect of this lower bound and Appendix A.2 for the derivations on the equivalence between the inequality for linear programming and equality in Eqs. (10,11,12,13,14) for optimal transport.

We define the marginal distributions for row $\mathbf{r} = [\mathbf{1}_N^T; N(\mathbf{w}_+^T \mathbf{1}_k - \rho \mathbf{w}_-^T \mathbf{1}_K)]^T \in \mathbb{R}^{N+1}$ and column $\mathbf{c} = [\mathbf{w}_+ N; N(1 - \rho \mathbf{w}_-^T \mathbf{1}_K)]^T \in \mathbb{R}^{K+1}$. We define the prediction matrix over the unlabeled set, which satisfied the confidence test, $P \in \mathbb{R}^{M \times N \times K}$ such that each element $P_{m,i,k} = p_m(y = k \mid \mathbf{x}_i)$ and the averaging prediction over $M$ models is $\bar{P} = \frac{1}{M} \sum_{m=1}^{M} P_{m,*,*} \in \mathbb{R}^{N \times K}$. The cost and the assignment matrices in $\mathbb{R}^{(N+1) \times (K+1)}$ are defined below

$$C := \begin{bmatrix} -\log \bar{P} & \mathbf{0}_N \\ \mathbf{0}_K^T & 0 \end{bmatrix} \tag{15} \qquad\qquad \tilde{Q} := \begin{bmatrix} Q & \mathbf{u} \\ \mathbf{v}^T & \tau \end{bmatrix}. \tag{16}$$

We derive in Appendix A.3 the optimization process to learn $\tilde{Q}$. After estimating $\tilde{Q}$, we get $Q$ from Eq. (16) and assign labels to unlabeled data, i.e., where $Q_{i,k} > 0$, and repeat the whole pseudo-labeling process for a few iterations as shown in Algorithm 1. The optimization process above can also be estimated using mini-batches (Fatras et al., 2020) if needed.

---

**Algorithm 1** Confident Sinkhorn Label Allocation (simplified)

---

**Input:** lab data $\{\mathbf{X}_l, \mathbf{y}_l\}$, unlab data $\mathbf{X}_u$, Sinkhorn reg $\varepsilon > 0$, fraction of assigned label $\rho \in [0, 1]$
**Output:** Allocation matrix $Q \in \mathbb{R}^{N \times K}$

1  Empirically estimate the label frequency $\mathbf{w}_-$ and $\mathbf{w}_+$ from $\{\mathbf{X}_l, \mathbf{y}_l\}$
2  **for** $t = 1, ..., T$ **do**
3      Train $M$ models $\theta_1, ..., \theta_M$ given the limited labeled data $\mathbf{X}_l, \mathbf{y}_l$
4      Obtain a confidence set $\mathbf{X}'_U \subset \mathbf{X}_U$ using T-value from Eq. (3) where $N := |\mathbf{X}'_U|$
5      Define a cost matrix $C = -\log p(y = [1, ..., K] \mid \mathbf{X}'_U)$
6      Set marginal distributions $\mathbf{r} = [\mathbf{1}_N^T; N(\mathbf{w}_+^T \mathbf{1}_K - \rho \mathbf{w}_-^T \mathbf{1}_K)]$ and $\mathbf{c} = [\mathbf{w}_+ N; N(1 - \rho \mathbf{w}_-^T \mathbf{1}_K)]$
    `/* Sinkhorn's algorithm                                                *`/
7      Initialize $\mathbf{a}^{(1)} = \mathbf{1}_K^T$
8      **while** $j = 1, ..., J$ *or until converged* **do**
9          Update $\mathbf{b}^{(j+1)} = \dfrac{\mathbf{r}}{\exp(-C_{i,k}/\varepsilon) \mathbf{a}^{(j)}}$   and   $\mathbf{a}^{(j+1)} = \dfrac{\mathbf{c}}{\exp(-C_{i,k}/\varepsilon)^T \mathbf{b}^{(j+1)}}$
10     Obtain $Q = \text{diag}(\mathbf{a}^{(J)}) \exp(-C_{i,k}/\varepsilon) \text{diag}(\mathbf{b}^{(J)})$      `// the assignment matrix`
11     Augment the assigned labels to $\{\mathbf{X}_l, \mathbf{y}_l\}$ and remove these points from the unlabeled data $\mathbf{X}_u$.

---

Instead of performing greedy selection using a threshold $\gamma$ like other PL algorithms, our CSA specifies the frequency of assigned labels including the lower bound $\mathbf{w}_-$ and upper bound $\mathbf{w}_+$ per class as well as the fraction of data points $\rho \in (0, 1)$ to be assigned. Then, the optimal transport will automatically perform row and column scalings to find the 'optimal' assignments such that the selected element, $Q_{i,k} > 0$, is among the highest values in the row (within a data point $i$-th) and at the same time receive the highest values in the column (within a class $k$-th). In contrast, existing PL will either look at the highest values in row or column, separately – but not jointly like ours. Here, the high value refers to high probability $P$ or low-cost $C$ in Eq. (4).

## 2.4 IPM PAC-BAYES BOUND FOR ENSEMBLE MODELS

In pseudo-labeling mechanisms, we first train a classifier on the labeled data, then use it to make predictions on the unlabelled for assigning pseudo-labels. Thus, it is important to guarantee the performance of the above process.

Our setting considers ensembling using multiple classifiers. We derive a PAC-Bayes bound to study the generalization ability of the proposed algorithm. Existing PAC-Bayes bounds for ensemble models has considered the Kullback-Leibler (KL) divergence (Masegosa et al., 2020) which becomes vacuous when using a finite number of models due to the absolute continuity requirement, which is rather concerning given our setting described above. Therefore, we complement and improve the existing result by developing the *first* PAC-Bayes result using Integral Probability Metrics (IPMs) Müller (1997) for ensemble models by utilizing a recent IPM PAC-Bayes result Amit et al. (2022).

Let $\Theta$ denote the set of all models and $\mathscr{P}(\Theta)$ be the distributions over $\Theta$. The IPM, for a function class $\mathcal{F}$ mapping from $\Theta \to \mathbb{R}$ is $\gamma_{\mathcal{F}}(p, q) = \sup_{f \in \mathcal{F}} |\mathbb{E}_p[f] - \mathbb{E}_q[f]|$ for $p, q \in \mathscr{P}(\Theta)$. Let $\xi \in \mathscr{P}(\Theta)$ then we define the *major voting* (MV) classifier as

$$\text{MV}_\xi(\mathbf{x}) := \arg\max_{y \in \mathcal{Y}} \mathbb{P}_{\theta \sim \xi}[h_\theta(\mathbf{x}) = y]. \tag{17}$$

We borrow notation from Masegosa et al. (2020) and define the *empirical tandem loss* where $h_\theta$ corresponds to the individual prediction of model $\theta$:

$$\hat{L}(\theta, \theta', \mathcal{D}_l) := \frac{1}{N_l} \sum_{(\mathbf{x}, y) \in \mathcal{D}_l} \mathbb{1}\Big[h_\theta(\mathbf{x}) = y\Big] \cdot \mathbb{1}\Big[h_{\theta'}(\mathbf{x}) = y\Big].$$

We introduce standard notation in PAC-Bayes such as the generalization gap for a loss $\ell : \Theta \times \mathcal{X} \times \mathcal{Y} \to [0, 1]$: $\Delta_{\mathcal{D}}(\theta) := \mathbb{E}_{(\mathbf{x}, y) \sim P}[\ell(\theta, \mathbf{x}, y)] - \mathbb{E}_{(\mathbf{x}, y) \in \mathcal{D}}[\ell(\theta, \mathbf{x}, y)]$, where $P$ is the population distribution. We are now ready to present the main Theorem when $\ell$ is taken to the $0 - 1$ loss.

Table 2: Comparison with pseudo-labeling methods on classification problem.

| Datasets | Supervised Learning | | Vime | PL | Flex | UPS | SLA | CSA |
|---|---|---|---|---|---|---|---|---|
| | XGBoost | MLP | | | | | | |
| Segment | $95.42 \pm 1$ | $94.63 \pm 1$ | $92.71 \pm 1$ | $95.68 \pm 1$ | $95.68 \pm 1$ | $95.67 \pm 1$ | $95.80 \pm 1$ | $\mathbf{95.90} \pm 1$ |
| Wdbc | $89.20 \pm 3$ | $91.33 \pm 2$ | $\mathbf{91.83} \pm 5$ | $91.23 \pm 3$ | $91.23 \pm 3$ | $91.62 \pm 3$ | $90.61 \pm 2$ | $\mathbf{91.83} \pm 3$ |
| Analcatdata | $91.63 \pm 2$ | $96.17 \pm 1$ | $96.13 \pm 2$ | $90.95 \pm 2$ | $90.62 \pm 3$ | $91.33 \pm 3$ | $90.98 \pm 2$ | $\mathbf{96.60} \pm 2$ |
| German-credit | $70.73 \pm 2$ | $71.10 \pm 3$ | $70.50 \pm 4$ | $70.72 \pm 3$ | $70.72 \pm 3$ | $71.15 \pm 2$ | $70.72 \pm 3$ | $\mathbf{71.47} \pm 3$ |
| Madelon | $54.80 \pm 3$ | $50.80 \pm 2$ | $52.97 \pm 2$ | $56.45 \pm 4$ | $56.74 \pm 4$ | $57.13 \pm 3$ | $55.13 \pm 3$ | $\mathbf{57.51} \pm 3$ |
| Dna | $88.53 \pm 1$ | $76.50 \pm 2$ | $79.07 \pm 3$ | $88.17 \pm 1$ | $88.17 \pm 1$ | $88.51 \pm 1$ | $88.09 \pm 2$ | $\mathbf{89.24} \pm 1$ |
| Agar Lepiota | $57.63 \pm 1$ | $63.80 \pm 1$ | $\mathbf{63.83} \pm 1$ | $58.98 \pm 1$ | $59.53 \pm 1$ | $58.88 \pm 1$ | $58.96 \pm 1$ | $59.53 \pm 1$ |
| Breast cancer | $93.20 \pm 2$ | $86.40 \pm 6$ | $85.87 \pm 5$ | $92.89 \pm 2$ | $92.89 \pm 2$ | $93.38 \pm 2$ | $92.76 \pm 2$ | $\mathbf{93.55} \pm 2$ |
| Digits | $82.23 \pm 3$ | $86.80 \pm 1$ | $84.10 \pm 1$ | $81.67 \pm 3$ | $81.44 \pm 3$ | $83.78 \pm 3$ | $81.51 \pm 3$ | $\mathbf{88.10} \pm 2$ |

**Theorem 2.** *Let $\mathcal{F}$ be a set of bounded and measurable functions. For any fixed labeled dataset $\mathcal{D}_l$ of size $N_l$, suppose $2(N_l - 1)\Delta^2_{\mathcal{D}_l} \in \mathcal{F}$ then for any $\delta \in (0,1)$, prior $\pi \in \mathscr{P}(\Theta)$ and $\xi \in \mathscr{P}(\Theta)$ it holds*

$$\mathbb{E}_{(\boldsymbol{x},y) \sim P}\left[\mathbb{1}\left[\mathrm{MV}_{\xi}(\boldsymbol{x}) = y\right]\right] \leq \mathbb{E}_{\xi}\left[\frac{4}{N_l}\sum_{(\boldsymbol{x},y) \in \mathcal{D}_l}\mathbb{1}\left[h_{\theta}(\boldsymbol{x}) = y\right]\right] + \sqrt{\frac{8\gamma_{\mathcal{F}}(\xi,\pi) + 8\ln(N_l/\delta)}{N_l - 1}},$$

*with probability at least $1 - \delta$.*

The above result allows us to reason about the generalization performance of the ensemble classifier trained on labeled set and used to predict on unlabelled set. Particularly, we derive the result using the IPM, instead of KL divergence in existing work Masegosa et al. (2020). Note, as an extreme example, we can select $\mathcal{F}$ to be the set of all measurable and bounded functions into $[-1, 1]$ and $\gamma_{\mathcal{F}}(p, q) = \mathrm{TV}(p, q)$ becomes the Total Variation (TV) distance. In this case, a simple application of Pinsker's inequality Csiszár and Körner (2011), which states that $\gamma_{\mathcal{F}}(p, q) \leq \sqrt{0.5\,\mathrm{KL}(p, q)}$, recovers a bound similar to, but tighter than, that of Masegosa et al. (2020) by a square root factor, see Eq. (4) in Theorem 8 of Masegosa et al. (2020). Therefore, the result we present is tighter and much more general since $\mathcal{F}$ can be chosen to be much smaller, as long as the assumptions are met. Our PAC-Bayes result also provides a useful theoretical analysis which may be of interest from the wider machine learning community using ensemble models.

## 3 EXPERIMENTS

### 3.1 PSEUDO-LABELING FOR CLASSIFICATION

**Setting.** In the first set of experiment, we select to compare with the following pseudo-labeling methods: Pseudo-labeling (PL) (Lee et al., 2013), FlexMatch (Zhang et al., 2021), UPS (Rizve et al., 2021) and SLA (Tai et al., 2021). We adapt the label assignment mechanisms in Zhang et al. (2021); Rizve et al. (2021) for tabular domains although their original designs are for computer vision tasks. Without explicitly stated, we use a pre-defined threshold of 0.8 for PL, FlexMatch, UPS. We further vary this threshold with different values in Sec. 3.3. We also compare with the supervised learning method – trained using only the labeled data. The multi-layer perceptron (MLP) implemented in Yoon et al. (2020) includes two layers using 100 hidden units. Additional to these pseudo-labeling based approaches, we compare the proposed method with Vime.

**Data.** We use public datasets for the classification task from UCI repository (Asuncion and Newman, 2007). Without using domain assumption, these data are presented as vectorized or tabular data formats. We also use Cifar10 and Cifar100 for the experiments with deep learning model. We summarize all dataset statistics in Appendix Table 9. Our data covers various domains including: image, language, medical, biology, audio, finance and retail.

**Classifier and hyperparameters.** Given limited labeled data presented in tabular format, we choose XGBoost (Chen and Guestrin, 2016) as the main classifier which typically outperforms state-of-the-art deep learning approaches in this setting (Shwartz-Ziv and Armon, 2022) although our method is more general and not restricted to XGBoost. We use $M = 20$ XGBoost models for ensembling. We refer to Appendix B.3 on the empirical analysis with different choices of $M$. The ranges for these hyperparameters are summarized in Appendix Table 7. We repeat the experiments 30 times with different random seeds.

Table 3: Classification accuracy on Cifar datasets.

| Methods | Cifar10 | | Cifar100 | |
|---|---|---|---|---|
| | 1000 labels | 4000 labels | 4000 labels | 10,000 labels |
| SL | $72.34 \pm 2.1$ | $83.35 \pm 2.3$ | $48.51 \pm 3.8$ | $60.29 \pm 3.3$ |
| PL (Lee et al., 2013) | $77.40 \pm 1.7$ | $87.06 \pm 1.9$ | $57.14 \pm 2.9$ | $66.53 \pm 2.5$ |
| Flex (Zhang et al., 2021) | $90.22 \pm 1.6$ | $91.89 \pm 2.4$ | $56.54 \pm 2.1$ | $67.62 \pm 1.3$ |
| SLA (Tai et al., 2021) | $91.44 \pm 1.2$ | $93.87 \pm 1.3$ | $56.92 \pm 2.3$ | $68.03 \pm 1.0$ |
| UPS (Rizve et al., 2021) | $91.86 \pm 0.7$ | $93.64 \pm 0.7$ | $\mathbf{59.23} \pm 1.2$ | $67.89 \pm 1.1$ |
| CSA | $\mathbf{92.38} \pm 0.8$ | $\mathbf{94.21} \pm 0.7$ | $58.62 \pm 1.3$ | $\mathbf{68.79} \pm 0.9$ |

**Result.** We use accuracy as the main metric for evaluating the classification problem (Wu and Zhou, 2017). We first compare our CSA with the methods in pseudo-labeling family. We show the performance achieved on the held-out test set as we iteratively assign pseudo-label to the unlabeled samples, i.e., varying $t = 1, ..., T$. As presented in Fig. 3 and Appendix Fig. 10, CSA significantly outperforms the baselines by a wide margin in the datasets of Analcatdata, Synthetic Control and Digits. CSA improves approximately 6% compared with fully supervised learning. CSA gains 2% in Madelon dataset and in the range of $[0.4\%, 0.8\%]$ on other datasets.

Unlike other pseudo-labeling methods (PL, FlexMatch, UPS), CSA can get rid of the requirement to predefine the suitable threshold $\gamma \in [0, 1]$, which is unknown in advance and should vary with different datasets. In addition, the selection in CSA is non-greedy that considers the relative importance within and across rows and columns by the Sinkhorn's algorithm while the existing PL methods can be greedy when only compared against a predefined threshold.

Furthermore, our CSA outperforms SLA (Tai et al., 2021), which does not take into consideration the confidence score. Thus, SLA may assign labels to 'uncertain' samples which can degrade the performance. From the comparison with SLA, we have shown that uncertainty estimation provides us a tool to identify susceptible points to be excluded and thus can be beneficial for SSL.

We then compare our CSA with Vime (Yoon et al., 2020), the current state-of-the-art methods in semi-supervised learning for tabular data in Table 2. CSA performs better than Vime in most cases, except the Agaricus Lepiota dataset which has an adequately large size (6500 samples, see Table 9) – thus the deep network component in Vime can successfully learn and perform well.

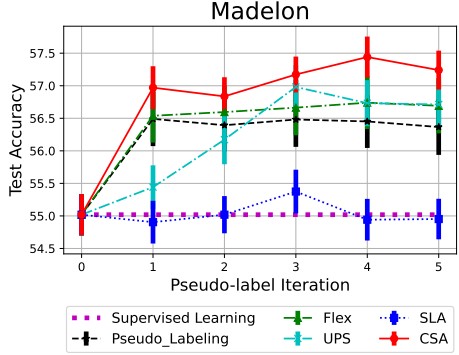

Figure 3: Comparison with PL methods

### 3.2 PSEUDO-LABELING WITH DEEP LEARNING MODEL

We further integrate our CSA strategy into a deep learning model for enabling large-scale experiments. In particular, we follow the same settings in Rizve et al. (2021) on Cifar10 for two different labeled set sizes (1000 and 4000 labels) as well as on CIFAR100 (with 4000 and 10,000 labels). The result is reported over 5 independent runs.

We follow the official implementation of UPS[3] to make a fair comparison. All models use the same backbone networks (Wide ResNet 28-2 (Zagoruyko and Komodakis, 2016)). Both CSA and UPS use the same MC-Dropout (Gal and Ghahramani, 2016) to estimate the uncertainty. But then CSA uses the T-test to filter out the samples. The result in Table 3 is consistent with the result in Table 5 of Rizve et al. (2021). Note that the implementation in Rizve et al. (2021) does not take into account the data augmentation step. Thus, the result of SLA is slightly lower than in Tai et al. (2021). It should be noted that our focus in this comparison is the effectiveness of pseudo-labeling strategies, not the data augmentation strategies for computer vision.

### 3.3 COMPARISON AGAINST PL WITH DIFFERENT THRESHOLDS

One of the key advantages of using OT is to learn the non-greedy assignment by performing row-scaling and column-scaling to get the best assignment such that the cost $C$ is minimized, or the

---
[3] https://github.com/nayeemrizve/ups

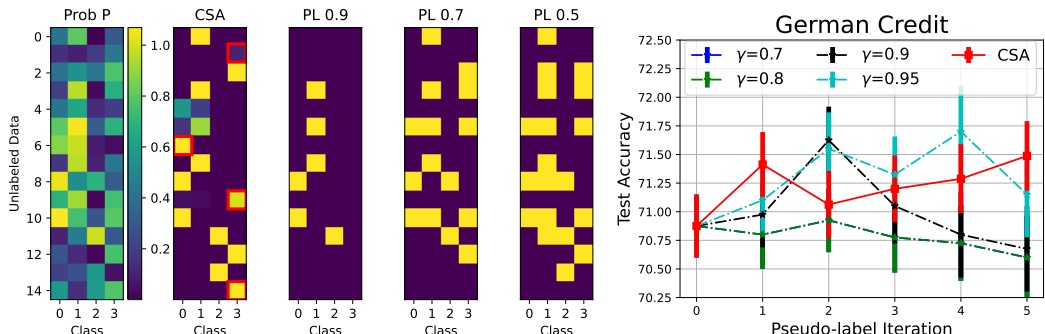

Figure 4: Comparison between CSA versus pseudo-labeling with different thresholds $\gamma$. *Left*: the assignments by CSA can not be simply achieved by varying the threshold in PL, *eg.* see the locations highlighted in red square. *Right*: Comparison when varying thresholds in PL against CSA.

likelihood is maximized with respect to the given constraints from the row **r** and column **c** marginal distributions. By doing this, our CSA can get rid of the necessity to define the threshold $\gamma$ and perform greedy selection in most of the existing pseudo-labeling methods.

We may be wondering if the optimal transport assignment $Q$ can be achieved by simply varying the thresholds $\gamma$ in pseudo-labeling methods? The answer is no. To back up our claim, we perform two analyses. First, we visualize and compare the assignment matrix achieved by CSA and by varying a threshold $\gamma \in \{0.5, 0.7, 0.9\}$ in PL. As shown in *left* Fig. 4, the outputs are different and there are some assignments which could not be selected by PL with varying $\gamma$, *eg.*, the following {row index, column index}: $\{1,3\}$, $\{9,3\}$ and $\{14,3\}$ annotated in red square in *left* Fig. 4.

Second, we empirically compare the performance of CSA against varying $\gamma \in \{0.7, 0.8, 0.9, 0.95\}$ in PL. We present the results in *right* Fig. 4, which again validates our claim that the optimal assignment in CSA will lead to better performance consistently than PL with changing values of $\gamma$.

## 3.4 ABLATION STUDIES

**Varying the number of labels and unlabels.** Different approaches exhibit substantially different levels of sensitivity to the amount of labeled and unlabeled data, as mentioned in Oliver et al. (2018). We, therefore, validate our claim in Theorem 1 by empirically demonstrating the model performance with increasing the number of unlabeled examples. In *left* Fig. 5, we show that the model will be beneficial with increasing the number of unlabeled data points.

**Limitation.** In some situations, such as when labeled data is too small in size or contains outliers, our CSA will likely assign incorrect labels to the unlabeled points at the earlier iteration. This erroneous will be accumulated further and lead to poor performance, as shown in *left* Fig. 5 when the number of labels is 100 and the number of unlabels is less than 500. We note, however, that other pseudo-labeling methods will share the same limitation.

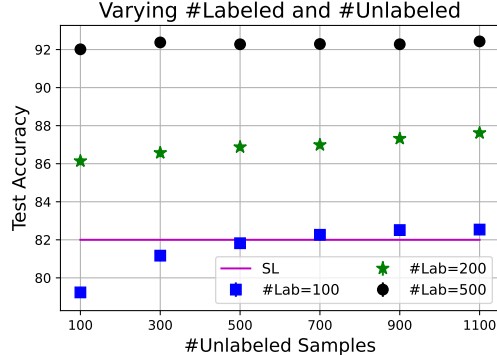

Figure 5: Performance on Digit w.r.t. the number of unlabeled samples given the number of labeled samples as 100, 200, 500, respectively.

**Further analysis.** We refer to Appendix B for other empirical analysis which can be sensitive to the performance: (i) different choices of PL iterations $T$ and (ii) different number of choices for XGB models $M$, (iii) computational time for each component and (iv) statistics on the number of points selected by each component per iteration and (v) effect of ensembling toward the final performance.

# 4 CONCLUSION AND FUTURE WORK

We have presented new theoretical results for pseudo-labeling, characterizing several interesting aspects to obtain good performance, such as the number of unlabels, the number of classes and in particular the uncertainty. Additionally, we derive the first PAC-Bayes generalization bound for ensemble models using IPM which extends and complements the existing result with KL divergence.

Based on the theoretical finding on the role of uncertainty, we have proposed the CSA which estimates and takes into account the confidence levels in assigning labels. The label assignment strategy is estimated using Sinkhorn algorithm which appropriately scales rows and columns probability to get assignment without the greedy selection using a threshold as used in most of PL methods.

The proposed CSA maintains the benefits of PL in its simplicity, generality, and ease of implementation while CSA can significantly improve PL performance by addressing the overconfidence issue and better assignment with OT. Our CSA can also be integrated into training deep learning model for large-scale experiments.

Future work can extend the CSA beyond semi-supervised learning tasks, such as to active learning or learning with noisy labels where pseudo-labeling is particularly useful.

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

Table 4: Notations. We use column vectors in all notations.

| Variable | Domain | Meaning |
|---|---|---|
| $K$ | $\mathcal{N}$ | number of classes |
| $M$ | $\mathcal{N}$ | number of XGBoost models for ensembling |
| $d$ | $\mathcal{N}$ | feature dimension |
| $\mathbf{x}_i$ | $\mathbb{R}^d$ | a data sample and a collection of them $\mathbf{X} = [\mathbf{x}_1, ..., \mathbf{X}_N] \in \mathbb{R}^{N \times d}$ |
| $y_i$ | $\{1, 2, ..., K\}$ | a data point label and a collection of labels $\mathbf{Y} = [y_1, ... y_N] \in \mathbb{R}^N$ |
| $\mathcal{D}_l, \mathcal{D}_u$ | sets | a collection of labeled $\{\mathbf{x}_i, y_i\}_{i=1}^{N_l}$ or unlabeled data $\{\mathbf{x}_i\}_{i=N_l+1}^{N_u+N_l}$ |
| $\mathbf{1}_K, \mathbf{1}_N$ | $[1, ..., 1]^T$ | a vector of ones in $K$ (or $N$) dimensions |
| $\rho$ | $[0, 1]$ | allocation fraction, $\rho = 0$ is no allocation or $\rho = 1$ is full allocation |
| $\varepsilon$ | $\mathbb{R}^+$ | Sinkhorn regularization parameter |
| $Q$ | $\mathbb{R}^{N \times K}$ | assignment matrix, i.e., $Q_{i,k} > 0$ assigns a label $k$ to a data point $i$ |
| $C$ | $\mathbb{R}^{N \times K}$ | a cost matrix estimated from the probability matrix i.e., $C = 1 - P$ |
| $\mathbf{w}, \mathbf{w}_+, \mathbf{w}_-$ | vector $[0, 1]^K$ | label frequency, upper bound and lower bound of label frequency |
| $\gamma$ | $[0, 1]$ | a threshold for assigning labels used in pseudo-labeling methods |

# A    APPENDIX: THEORETICAL RESULTS

We derive the proof for Theorem 1 in Sec. A.1, interpret the linear programing inequality to optimal transport in Sec. A.2, present the derivations for the Sinkhorn algorithm used in our method in Sec. A.3, the proof for Theorem 2 in Sec. A.4 and consider another criterion for confident estimation in Sec. B.1.

## A.1    PROOF OF THEOREM 1 IN THE MAIN PAPER

We consider a multi-classification problem with the generating feature $\mathbf{x} \mid y = k \overset{\text{iid}}{\sim} \mathcal{N}(\mu_k, \Lambda)$ where $\mu_k \in \mathbb{R}^d$ is the means of the $k$-th classes, $\Lambda$ is the covariance matrix defining the variation of the data noisiness across input dimensions. For simplicity in the analysis, we define $\Lambda = \text{diag}([\sigma_1^2, ....\sigma_d^2])$ as a $\mathbb{R}^{d \times d}$ diagonal matrix which does not change with different classes $k$.

Let $\{\tilde{X}_i^k\}_{i=1}^{\tilde{n}_k}$ be the set of unlabeled data whose pseudo-label is $k$, respectively. Let $\{I_i^k\}_{i=1}^{\tilde{n}_k}$ be the binary indicator of correct assignments, such as if $I_i^k = 1$, then $\tilde{X}_i^k \sim \mathcal{N}(\mu_k, \Lambda)$.

We assume the binary indicators $I_i^k$ to be generated from a pseudo-labeling classifier with the corresponding expectations $\mathbb{E}(I^k)$ and variances of $\text{Var}(I^k)$ for the generating processes.

In uncertainty estimation literature [13; 21], $\Lambda$ is called the *aleatoric uncertainty* (from the observations) and $\text{Var}(I^+), \text{Var}(I^-)$ are the *epistemic uncertainty* (from the model knowledge).

Instead of using linear classifier in the binary case as in (Yang and Xu, 2020), here we consider a set of probabilistic classifiers: $f_k(\mathbf{x}_i) := \mathcal{N}(\mathbf{x}_i | \hat{\boldsymbol{\theta}}_k, \Lambda), \forall k = \{1, ..., K\}$ to classify $y_i = k$ such that $k = \arg\max_{j \in [1, ..., K]} f_j(\mathbf{x}_i)$. Given the above setup, it is natural to construct our estimate as $\hat{\theta}_k = \frac{\sum_{i=1}^{\tilde{n}_k} \tilde{X}_i^k}{\tilde{n}_k}$ via the extra unlabeled data. We aim to characterize the estimation error $\left|\sum_{k=1}^{K} \hat{\theta}_k - \mu_k\right|$.

**Theorem 3** (restated Theorem 1). *For $\delta > 0$, our estimate $\hat{\theta}_k$ for the above pseudo-label classification satisfies*

$$\sum_{k=1}^{K} \left|\hat{\theta}_k - \mu_k\right| \leq \delta \tag{18}$$

*with a probability at least* $1 - 2\sum_{k=1}^{K}\sum_{j=1}^{d}\exp\left\{-\frac{\delta^2 \tilde{n}_k}{8\sigma_j^2}\right\} - \sum_{k=1}^{K}\frac{4\text{Var}(I^k)}{\delta^2}\left|\mu_k - \mu_{\backslash k}\right|.$

*Proof.* Our proof extends the result from Yang and Xu (2020) in three folds: (i) generalizing from binary to multi classification; (ii) extending from one dimensional to multi-dimensional input; and (iii) taking into account the uncertainty for pseudo-labeling. Let us denote $\mu_{\backslash k} = \mu_j \mid \exists j \in \{1,...,K\} \backslash k$. Here, $\backslash k$ denotes one of the remaining class, except $k$. We rewrite $\tilde{X}_i^k = (1 - I_i^k)(\mu_{\backslash k} - \mu_k) + Z_i^k$ where $Z_i^k \sim \mathcal{N}(\mu_k, \sigma^2)$ and $I_i^k$ is generated from a classifier with the expectation $\mathbb{E}(I^k)$ and variance $\text{Var}(I_i^k)$. This means if the pseudo-label is correct, $\tilde{X}_i^k \sim Z_i^k$ and otherwise $\tilde{X}_i^k \sim (\mu_{\backslash k} - \mu_k) + Z_i^k$. We want to bound the estimation error $\sum_{k=1}^{K}\left|\hat{\theta}_k - \mu_k\right|$. Our estimate $\hat{\theta}_k$ can be written as

$$\hat{\theta}_k = \frac{\sum_{i=1}^{\tilde{n}_k}\tilde{X}_i^k}{\tilde{n}_k} = \frac{\sum_{i=1}^{\tilde{n}_k}(1 - I_i^k)(\mu_{\backslash k} - \mu_k) + Z_i^k}{\tilde{n}_k} \tag{19}$$

$$= \left(\frac{\sum_{i=1}^{\tilde{n}_k}I_i^k}{\tilde{n}_k}(\mu_k - \mu_{\backslash k}) + \frac{\sum_{i=1}^{\tilde{n}_k}Z_i^k}{\tilde{n}_k}\right). \tag{20}$$

Given $\sum_{k=1}^{K}\mathbb{E}(I^k)\left[\mu_k - \mu_{\backslash k}\right] = 0$, we write

$$\sum_{k=1}^{K}\left|\hat{\theta}_k - \mu_k\right| = \sum_{k=1}^{K}\left|\hat{\theta}_k - \mu_k\right| - \sum_{k=1}^{K}\mathbb{E}(I^k)\left[\mu_k - \mu_{\backslash k}\right]. \tag{21}$$

We continue by plugging Eq. (20) into Eq. (21) to obtain

$$\sum_{k=1}^{K}\left|\hat{\theta}_k - \mu_k\right| = \sum_{k=1}^{K}\left|\frac{\sum_{i=1}^{\tilde{n}_k}I_i^k}{\tilde{n}_k}(\mu_k - \mu_{\backslash k}) + \frac{\sum_{i=1}^{\tilde{n}_k}Z_i^k}{\tilde{n}_k} - \mu_k\right| - \sum_{k=1}^{K}\mathbb{E}(I^k)\left[\mu_k - \mu_{\backslash k}\right] \tag{22}$$

$$\leq \sum_{k=1}^{K}\left|\frac{\sum_{i=1}^{\tilde{n}_k}I_i^k}{\tilde{n}_k} - \mathbb{E}(I^k)\right|\left|\mu_k - \mu_{\backslash k}\right| + \sum_{k=1}^{K}\left|\frac{\sum_{i=1}^{\tilde{n}_k}Z_i^k}{\tilde{n}_k} - \mu_k\right|. \tag{23}$$

We have used the triangle inequality to obtain Eq. (23). Next we bound each term in orange and blue separately.

First, we bound $\frac{\sum_{i=1}^{\tilde{n}_k}I_i^k}{\tilde{n}_k}$ using Chebyshev's inequality as follows

$$P\left(\left|\frac{\sum_{i=1}^{\tilde{n}_k}I_i^k}{\tilde{n}_k} - \mathbb{E}(I^k)\right| > t\right) \leq \frac{\text{Var}(I^k)}{t^2}. \tag{24}$$

Using the union bound across $K$ classes, we get

$$P\left(\bigcup_{k=\{1...K\}}\left\{\left|\frac{\sum_{i=1}^{\tilde{n}_k}I_i^k}{\tilde{n}_k} - \mathbb{E}(I^k)\right| > t\right\}\right) \leq \sum_{k=1}^{K}P\left(\left|\frac{\sum_{i=1}^{\tilde{n}_k}I_i^k}{\tilde{n}_k} - \mathbb{E}(I^k)\right| > t\right) \tag{25}$$

$$\leq \sum_{k=1}^{K}\frac{\text{Var}(I^k)}{t^2}. \tag{26}$$

Second, we bound the term

$$\frac{\sum_{i=1}^{\tilde{n}_k}Z_i^k}{\tilde{n}_k} \sim \mathcal{N}\left(\mu_k, \frac{\Lambda}{\tilde{n}_k}\right) = \mathcal{N}\left(\mu_k, \frac{\text{diag}[\sigma_1^2,...,\sigma_d^2]}{\tilde{n}_k}\right). \tag{27}$$

Due to the diagonal Gaussian distribution, we can apply Gaussian concentration to each dimension $j = \{1,...,d\}$ to have that

$$P\left(\left|\frac{\sum_{i=1}^{\tilde{n}_k}Z_i^{k,j}}{\tilde{n}_k} - \mu_{k,j}\right| > t\right) \leq 2\exp\left\{-\frac{t^2 \tilde{n}_k}{2\sigma_j^2}\right\}. \tag{28}$$

Using union bound, we get

$$P\left(\bigcup_{j=\{1...d\}}\left|\frac{\sum_{i=1}^{\tilde{n}_k}Z_i^{k,j}}{\tilde{n}_k}-\mu_{k,j}\right|>t\right)\leq\sum_{j=1}^d P\left(\left|\frac{\sum_{i=1}^{\tilde{n}_k}Z_i^{k,j}}{\tilde{n}_k}-\mu_{k,j}\right|>t\right) \tag{29}$$

$$\leq 2\sum_{j=1}^d\exp\left\{-\frac{t^2\tilde{n}_k}{2\sigma_j^2}\right\}. \tag{30}$$

We again apply union bound over $\bigcup_{k=\{1...K\}}$ to Eq. (30)

$$P\left(\bigcup_{k=\{1...K\}}\bigcup_{j=\{1...d\}}\left|\frac{\sum_{i=1}^{\tilde{n}_k}Z_i^{k,j}}{\tilde{n}_k}-\mu_{k,j}\right|>t\right)\leq\sum_{k=1}^K P\left(\bigcup_{j=\{1...d\}}\left|\frac{\sum_{i=1}^{\tilde{n}_k}Z_i^{k,j}}{\tilde{n}_k}-\mu_{k,j}\right|>t\right) \tag{31}$$

$$\leq 2\sum_{k=1}^K\sum_{j=1}^d\exp\left\{-\frac{t^2\tilde{n}_k}{2\sigma_j^2}\right\}. \tag{32}$$

Let $\delta>0$, we consider an event $E$ as follows

$$E=\left\{\bigcup_{k=\{1...K\}}\left|\frac{\sum_{i=1}^{\tilde{n}_k}I_i^k}{\tilde{n}_k}-\mathbb{E}(I^k)\right|\left|\mu_k-\mu_{\backslash k}\right|\leq\frac{\delta}{2}\quad and\quad\bigcup_{k=\{1...K\}}\left|\frac{\sum_{i=1}^{\tilde{n}_k}Z_i^k}{\tilde{n}_k}-\mu_k\right|\leq\frac{\delta}{2}\right\}. \tag{33}$$

Using the results from Eqs. (26, 32) and applying the union bound to Eq. (33), we get

$$P(E)\geq 1-2\sum_{k=1}^K\sum_{j=1}^d\exp\left\{-\frac{\delta^2\tilde{n}_k}{8\sigma_j^2}\right\}-\sum_{k=1}^K\frac{4\mathtt{Var}(I^k)}{\delta^2}\left|\mu_k-\mu_{\backslash k}\right|. \tag{34}$$

Equivalently, we write

$$P\left(\sum_{k=1}^K\left|\hat{\theta}_k-\mu_k\right|\leq\delta\right)\geq 1-2\sum_{k=1}^K\sum_{j=1}^d\exp\left\{-\frac{\delta^2\tilde{n}_k}{8\sigma_j^2}\right\}-\sum_{k=1}^K\frac{4\mathtt{Var}(I^k)}{\delta^2}\left|\mu_k-\mu_{\backslash k}\right|. \tag{35}$$

$\square$

## A.2 DERIVATION OF THE OPTIMAL TRANSPORT LINEAR PROGRAMMING

We start with the following linear programming inequality.

$$\text{minimize}_Q\quad\langle Q,C\rangle \tag{36}$$
$$\text{s.t.}\quad Q_{ik}\geq 0 \tag{37}$$
$$Q\mathbf{1}_K\leq\mathbf{1}_N \tag{38}$$
$$Q^T\mathbf{1}_N\leq N\mathbf{w}_+ \tag{39}$$
$$\mathbf{1}_N^TQ\mathbf{1}_K\geq N\rho\mathbf{w}_-^T\mathbf{1}_K \tag{40}$$

where $\mathbf{1}_K$ and $\mathbf{1}_N$ are the vectors one in $K$ and $N$ dimensions, respectively; $\rho\in[0,1]$ is the fraction of assigned label, i.e., $\rho=1$ is full allocation; $\mathbf{w}_+,\mathbf{w}_-\in\mathbb{R}^k$ are the vectors of upper and lower bound assignment per class which can be estimated empirically from the class label frequency in the training data or from prior knowledge.

Specifically, the row marginal $\mathbf{1}_N$ is a vector of one, i.e., each data point should not be assigned to more than one label, Eq. (38). The column marginal is in the range of $[N\mathbf{w}_-,N\mathbf{w}_+]$. Each column label should receive the amount of assignment ranging from lower bound and upper bound label frequency $[N\mathbf{w}_-,N\mathbf{w}_+]$, Eq. (39,40).

We then transform these above inequalities to optimal transport setup. As used in Tai et al. (2021), we introduce non-negative slack variables $u$, $v$ and $\tau$ to replace the inequality constraints on the marginal

distributions. We yield the following optimization problem:

$$\text{minimize}_{Q,\mathbf{u},\mathbf{v},\tau} \quad \langle Q, C \rangle \tag{41}$$

$$\text{s.t.} \quad Q_{ik} \geq 0, \mathbf{u} \succeq 0, \mathbf{v} \succeq 0, \tau \geq 0 \tag{42}$$

$$Q\mathbf{1}_K + \mathbf{u} = \mathbf{1}_N \tag{43}$$

$$Q^T \mathbf{1}_N + \mathbf{v} = \mathbf{w}_+ N \tag{44}$$

$$\mathbf{1}_N^T Q \mathbf{1}_K = \tau + N\rho \mathbf{w}_-^T \mathbf{1}_K \tag{45}$$

We substitute Eq. (40) into Eq. (43) to have:

$$\mathbf{u}^T \mathbf{1}_N + \tau = N(1 - \rho \mathbf{w}_-^T \mathbf{1}_K) \tag{46}$$

Similarly, we substitute Eq. (40) into Eq. (44) to obtain:

$$\mathbf{v}^T \mathbf{1}_K + \tau = N(\mathbf{w}_+^T \mathbf{1}_K - \rho \mathbf{w}_-^T \mathbf{1}_K) \tag{47}$$

We recognize the above equations as an optimal transport problem [11; 47] with row marginal distribution $\mathbf{r} = [\mathbf{1}_N^T; N(\mathbf{w}_+^T \mathbf{1}_K - \rho \mathbf{w}_-^T \mathbf{1}_K)]^T \in \mathbb{R}^{N+1}$ and column marginal distribution $\mathbf{c} = [\mathbf{w}_+ N; N(1 - \rho \mathbf{w}_-^T \mathbf{1}_K)]^T \in \mathbb{R}^{K+1}$.

### A.3 SINKHORN OPTIMIZATION STEP

Given the cost matrix $C \in \mathbb{R}^{N \times K}$, our original objective function is

$$\text{minimize}_{Q \in \mathbb{R}^{N \times K}} \langle Q, C \rangle \tag{48}$$

where $\langle Q, C \rangle := \sum_{ik} Q_{ik} C_{ik}$. For a general matrix $C$, the worst case complexity of computing that optimum scales in $\mathcal{O}\left(N^3 \log N\right)$ [11]. To overcome this expensive computation, we can utilize the entropic regularization to reduce the complexity to $\mathcal{O}\left(\frac{N^2}{\varepsilon^2}\right)$ [14] by writing the objective function as:

$$\mathcal{L}(Q) = \text{minimize}_{Q \in \mathbb{R}^{N \times K}} \langle Q, C \rangle - \varepsilon \mathbb{H}(Q) \tag{49}$$

where $\mathbb{H}(Q)$ is the entropy of $Q$. This is a strictly convex optimization problem [11] and has a unique optimal solution.

Since the solution $Q^*$ of the above problem is surely nonnegative, i.e., $Q_{i,k}^* > 0$, we can thus ignore the positivity constraints when introducing dual variables $\lambda^{(1)} \in \mathbb{R}^N, \lambda^{(2)} \in \mathbb{R}^K$ for each marginal constraint as follows:

$$\mathcal{L}(Q, \lambda^{(1)}, \lambda^{(2)}) = \langle Q, C \rangle + \varepsilon \mathbb{H}(Q) + \langle \lambda^{(1)}, \mathbf{r} - Q\mathbf{1}_K \rangle + \langle \lambda^{(2)}, \mathbf{c} - Q^T \mathbf{1}_N \rangle \tag{50}$$

where $\mathbf{r}$ and $\mathbf{c}$ are the row and column marginal distributions. We take partial derivatives w.r.t. each variable to solve the above objective. We have,

$$\frac{\partial \mathcal{L}(Q, \lambda^{(1)}, \lambda^{(2)})}{\partial Q_{i,k}} = C_{i,k} + \varepsilon \left( \log Q_{i,k} + 1 \right) - \lambda_i^{(1)} - \lambda_k^{(2)} = 0 \tag{51}$$

$$Q_{i,k}^* = \exp\left\{ \frac{\lambda_i^{(1)} + \lambda_k^{(2)} - C_{i,k}}{\varepsilon} + 1 \right\} \tag{52}$$

$$= \exp\left( \frac{\lambda_i^{(1)}}{\varepsilon} - 1/2 \right) \exp\left( -\frac{C_{i,k}}{\varepsilon} \right) \exp\left( \frac{\lambda_k^{(2)}}{\varepsilon} - 1/2 \right). \tag{53}$$

The factorization of the optimal solution in the above equation can be conveniently rewritten in matrix form as $\text{diag}(\mathbf{a}) \exp\left( -\frac{C_{i,k}}{\varepsilon} \right) \text{diag}(\mathbf{b})$ where $\mathbf{a}$ and $\mathbf{b}$ satisfy the following non-linear equations corresponding to the constraints

$$\text{diag}(\mathbf{a}) \exp\left( -\frac{C_{i,k}}{\varepsilon} \right) \text{diag}(\mathbf{b}) \mathbf{1}_K = \mathbf{r} \quad (54) \qquad \text{diag}(\mathbf{b}) \exp\left( -\frac{C_{i,k}}{\varepsilon} \right)^T \text{diag}(\mathbf{a}) \mathbf{1}_N = \mathbf{c}. \quad (55)$$

Now, instead of optimizing $\lambda^{(1)}$ and $\lambda^{(2)}$, we alternatively solve for the vectors

$$\mathbf{a} = \exp\left(\frac{\lambda^{(1)}}{\varepsilon} - 1/2\right) \qquad (56) \qquad\qquad \mathbf{b} = \exp\left(\frac{\lambda^{(2)}}{\varepsilon} - 1/2\right). \qquad (57)$$

The above forms the system of equations with two equations and two unknowns. We solve this system by initializing $\mathbf{b}^{(0)} = \mathbf{1}_K^T$ and iteratively updating for multiple iterations $j = 1...J$:

- Update $\mathbf{a}^{(j+1)} = \dfrac{\mathbf{r}}{\exp\left(-\frac{C_{i,k}}{\varepsilon}\right)\mathbf{b}^{(j)}}$

- Update $\mathbf{b}^{(j+1)} = \dfrac{\mathbf{c}}{\exp\left(-\frac{C_{i,k}}{\varepsilon}\right)^T \mathbf{a}^{(j+1)}}$

The division operator used above between two vectors is to be understood entry-wise. After $J$ iterations, we estimate the final assignment matrix for pseudo-labeling

$$Q^* = \text{diag}(\mathbf{a}^{(J)})\exp\left(-\frac{C_{i,k}}{\varepsilon}\right)\text{diag}(\mathbf{b}^{(J)}). \qquad (58)$$

## A.4 PROOF OF THEOREM 2

We begin by quoting Proposition 4 of Amit et al. (2022).

**Proposition A.1** (Amit et al. (2022))**.** *Let $\mathcal{F}$ be a set of bounded and measurable function. For any fixed dataset $\mathcal{D}$ of size $N$, suppose $2(N-1)\Delta_{\mathcal{D}}^2 \in \mathcal{F}$ then for any $\delta \in (0,1)$, prior $\pi \in \mathscr{P}(\Theta)$ and $\xi \in \mathscr{P}(\Theta)$ it holds*

$$\Delta_{\mathcal{D}}(\theta) \leq \sqrt{\frac{\gamma_{\mathcal{F}}(\xi,\pi) + \ln(N/\delta)}{2(N-1)}} \qquad (59)$$

We then require Theorem 3 from Masegosa et al. (2020).

**Theorem 4** (Masegosa et al. (2020))**.** *For any choice of $\xi$, we have*

$$\mathbb{E}_{(\boldsymbol{x},y)\sim P}\left[\mathbb{1}\left[\text{MV}_{\xi}(\boldsymbol{x}) = y\right]\right] \leq 4\mathbb{E}_{\xi(\theta)}\mathbb{E}_{\xi(\theta')}\left[\mathbb{E}_{(\boldsymbol{x},y)\sim P}\left[\mathbb{1}\left[h_{\theta}(\boldsymbol{x}) = y\right] \cdot \mathbb{1}\left[h_{\theta'}(\boldsymbol{x}) = y\right]\right]\right] \qquad (60)$$

**Theorem 5** (Theorem 2 in the main paper)**.** *Let $\mathcal{F}$ be a set of bounded and measurable functions. For any fixed labeled dataset $\mathcal{D}_l$ of size $N_l$, suppose $2(N_l-1)\Delta_{\mathcal{D}}^2 \in \mathcal{F}$ then for any $\delta \in (0,1)$, prior $\pi \in \mathscr{P}(\Theta)$ and $\xi \in \mathscr{P}(\Theta)$ it holds*

$$\mathbb{E}_{(\boldsymbol{x},y)\sim P}\left[\mathbb{1}\left[\text{MV}_{\xi}(\boldsymbol{x}) = y\right]\right] \leq \mathbb{E}_{\xi}\left[\frac{4}{N_l}\sum_{(\boldsymbol{x},y)\in\mathcal{D}_l}\mathbb{1}\left[h_{\theta}(\boldsymbol{x}) = y\right]\right] + \sqrt{\frac{8\gamma_{\mathcal{F}}(\xi,\pi) + 8\ln(N_l/\delta)}{N_l - 1}},$$

*with probability at least $1 - \delta$.*

*Proof.* By using Theorem 4, we have the below inequality

$$\mathbb{E}_{(\mathbf{x},y)\sim P}\left[\mathbb{1}\left[\text{MV}_{\xi}(\mathbf{x}) = y\right]\right] \leq 4\mathbb{E}_{\xi(\theta)}\mathbb{E}_{\xi(\theta')}\left[\mathbb{E}_{(\mathbf{x},y)\sim P}\left[\mathbb{1}\left[h_{\theta}(\mathbf{x}) = y\right] \cdot \mathbb{1}\left[h_{\theta'}(\mathbf{x}) = y\right]\right]\right] \qquad (61)$$

$$\leq 4\mathbb{E}_{\xi(\theta)}\left[\mathbb{E}_{(\mathbf{x},y)\sim P}\left[\mathbb{1}\left[h_{\theta}(\mathbf{x}) = y\right]\right]\right] \qquad (62)$$

$$\leq 4\mathbb{E}_{\xi}\left[\frac{1}{N_l}\sum_{(\mathbf{x},y)\in\mathcal{D}_l}\mathbb{1}\left[h_{\theta}(\mathbf{x}) = y\right]\right] + 4\Delta_{\mathcal{D}_l}(\xi) \qquad (63)$$

$$\leq 4\mathbb{E}_{\xi}\left[\frac{1}{N_l}\sum_{(\mathbf{x},y)\in\mathcal{D}_l}\mathbb{1}\left[h_{\theta}(\mathbf{x}) = y\right]\right] + \sqrt{\frac{8\gamma_{\mathcal{F}}(\xi,\pi) + 8\ln(N_l/\delta)}{N_l - 1}}, \qquad (64)$$

with probability $1 - \delta$. In Eq. (63), we have used the definition of $\Delta_{\mathcal{D}}(\theta) := \mathbb{E}_{(\mathbf{x},y)\sim P}[\ell(\theta,x,y)] - \mathbb{E}_{(\mathbf{x},y)\in\mathcal{D}}[\ell(\theta,x,y)]$ for a fixed $\theta' \in \Theta$ and used the Proposition A.1 for Eq. (64). This concludes our proof. $\qquad\square$

Table 5: Comparison with different choices for estimating the confidence scores. The performance with total variance reflects the uncertainty choice used in UPS (Rizve et al., 2021). CSA using T-test achieves the best performance. We also highlight that using CSA (either with Total variance of T-test) will consistently outperform SLA.

| Datasets | SLA | CSA Entropy | CSA Total variance | CSA T-test |
|---|---|---|---|---|
| Segment | $95.82 \pm 1$ | $95.74\,(1.0)$ | $95.70 \pm 1$ | $\mathbf{95.90} \pm 1$ |
| Wdbc | $90.64 \pm 3$ | $89.50(4.3)$ | $91.62 \pm 3$ | $\mathbf{91.83} \pm 3$ |
| Analcatdata | $90.43 \pm 3$ | $92.00 \pm 3$ | $92.40 \pm 2$ | $\mathbf{94.79} \pm 2$ |
| German-credit | $70.79 \pm 3$ | $70.88(3.0)$ | $71.75(2.8)$ | $\mathbf{71.47} \pm 3$ |
| Madelon | $55.96 \pm 3$ | $57.50\,(2.5)$ | $55.56\,(3.8)$ | $57.51 \pm 2$ |
| Dna | $87.86 \pm 2$ | $88.23\,(1.6)$ | $87.12(2.2)$ | $\mathbf{89.24} \pm 1$ |
| Agaricus Lepiota | $59.01 \pm 1$ | $57.67(1.0)$ | $59.59(1.0)$ | $59.53 \pm 1$ |
| Breast cancer | $92.65 \pm 2$ | $93.33(2.8)$ | $93.25(2.6)$ | $\mathbf{93.55} \pm 2$ |
| Digits | $81.03 \pm 3$ | $84.65(3.1)$ | $84.37(3.1)$ | $\mathbf{86.31} \pm 3$ |

## A.5 REPRESENTING THE TWO EMPIRICAL DISTRIBUTIONS FOR T-TEST

We train XGB as the main classifier using different sets of hyperparameters. Let denote the predictive probability using $M$ classifiers across $N$ unlabeled data points over $K$ classes as $P \in \mathbb{R}^{M \times N \times K}$. Given a data point $i$, we define the highest score and second-highest score classes as follows: $\diamond = \arg\max_{k=\{1...K\}} \frac{1}{M} \sum_{m=0}^{M} P_{m,i,k}$ and $\oslash = \arg\max_{k=\{1...K\} \backslash \diamond} \frac{1}{M} \sum_{m=0}^{M} P_{m,i,k}$ where $\{1...K\} \backslash \diamond$ means that we exclude an index $\diamond$ from a set $\{1...K\}$. Note that these indices of the highest $\diamond$ and second-highest score class $\oslash$ vary with different data points and thus can also be defined as the function $\diamond(i)$ and $\oslash(i)$.

We consider a predictive probability of a data point $i = 0$, i.e., $P_{M,i=0,K} \in \mathbb{R}^{M \times K}$. We then define two empirical distributions of predicting the highest $\mathcal{N}(\mu_{i,\diamond}, \sigma_{i,\diamond}^2)$ and second-highest class $\mathcal{N}(\mu_{i,\oslash}, \sigma_{i,\oslash}^2)$, such as the empirical means are:

$$\mu_{i,\diamond} = \frac{1}{M} \sum_{m=1}^{M} p_m(y = \diamond \mid \mathbf{x}_i) \qquad (65) \qquad \mu_{i,\oslash} = \frac{1}{M} \sum_{m=1}^{M} p_m(y = \oslash \mid \mathbf{x}_i) \qquad (66)$$

and the variances are defined respectively as

$$\sigma_{i,\diamond}^2 = \frac{1}{M} \sum_{m=1}^{M} \left( p_m(y = \diamond \mid \mathbf{x}_i) - \mu_{i,\diamond} \right)^2 \quad (67) \qquad \sigma_{i,\oslash}^2 = \frac{1}{M} \sum_{m=1}^{M} \left( p_m(y = \oslash \mid \mathbf{x}_i) - \mu_{i,\oslash} \right)^2. \quad (68)$$

# B APPENDIX: ADDITIONAL ABLATION STUDIES

## B.1 EXPERIMENTS WITH DIFFERENT UNCERTAINTY ESTIMATION CHOICES

**Total variance.** In multilabel classification settings [23], multiple high score classes can be considered together. Thus, the T-test between the highest and second-highest is no longer applicable. We consider the following total variance across classes as the second criteria. This total variance for uncertainty estimation has also been used in UPS [43] using Monte Carlo dropout [18] in the context of deep neural network:

$$\mathcal{V}[p(y \mid \mathbf{x})] \approx \frac{1}{K} \sum_{k=1}^{K} \Big[ \underbrace{\frac{1}{M} \sum_{m=1}^{M} \Big( p_{mk} - \sum_{m=1}^{M} \frac{p_{mk}}{M} \Big)^2}_{\text{variance of assigning } \mathbf{x} \text{ to class } k} \Big] \qquad (69)$$

where $p_{mk} := p_m(y = k \mid \mathbf{x})$ for brevity. Data points with high uncertainty measured by the total variance are excluded because a consensus of multiple classifiers is generally a good indicator of the labeling quality. In our setting, a high consensus is represented by low variance or high confidence. While the T-test naturally has a threshold of 2 to reject a data point, the total variance does not have such a threshold and thus we need to impose our own value. In the experiment, we reject 50% of the points with a high total variance score.

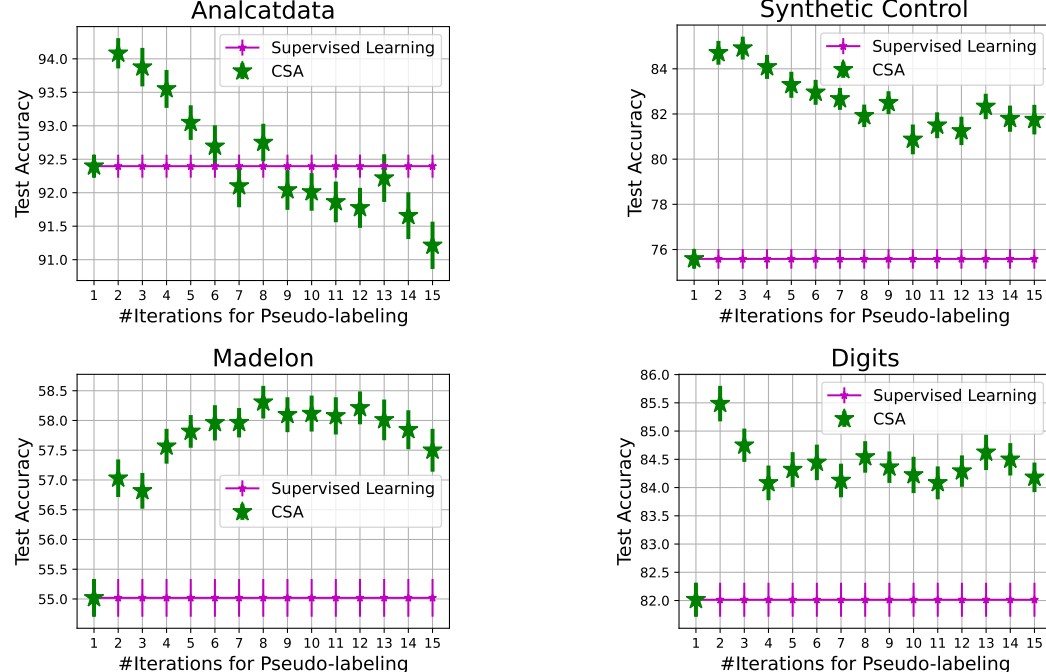

Figure 6: Performance with different choices of the iterations $T \in [1, ..., 15]$ for pseudo-labeling.

In Appendix Table 5, we empirically compare the performance using all of the different uncertainty choices, including the entropy criteria [32] (see Appendix B.1), and show that the Welch T-test is the best for this pseudo-labeling problem which can explicitly capture the uncertainty gap in predicting the highest and second-highest classes while total variance and entropy can not.

**Entropy.** We consider the *entropy* as another metric for estimating the confidence score for each data point. The entropy across multiple ensembles is as known as the total uncertainty (Malinin, 2019; Malinin et al., 2020), which is the sum of data and knowledge uncertainty. By considering the mutual information, we can estimate the *total uncertainty* as follows:

$$\mathbb{H}[p(y \mid \mathbf{x}, D)] \sim \mathbb{H}\left[\frac{1}{M}\sum_{m=1}^{M}\underbrace{p_m\Big(y=1,...,K \mid \mathbf{x}\Big)}_{\in \mathbb{R}^K}\right] \quad (70)$$

where $p_m(y \mid \mathbf{x}) := p(y \mid \mathbf{x}, \theta^{(m)})$ for brevity. We compare this total uncertainty against the proposed T-test and the total variance in Table 5. However, this entropy does not perform as well as the T-test and total variance. The possible explanation is that this total uncertainty can not well capture the variation in these class predictions, such as can not explicitly measure the gap between the highest and second-highest classes. Additionally in Table 5, we also show that the proposed T-test outperforms the total variance which is used in the existing UPS (Rizve et al., 2021). In addition, we highlight that using CSA (either with Total variance of T-test) will consistently outperform SLA.

## B.2 EXPERIMENTS WITH DIFFERENT NUMBER OF ITERATIONS $T$ FOR PSEUDO-LABELING

Pseudo-labeling is an iterative process by repeatedly augmenting the labeled data with selected samples from the unlabeled set. Under this repetitive mechanism, the error can also be accumulated during the PL process if the wrong assignment is made at an earlier stage. Therefore, not only the choice of pseudo-labeling threshold is important, but also the number of iterations $T$ used for the pseudo-labeling process can affect the final performance.

Therefore, we analyze the performance with respect to the choices of the number of iterations $T \in [1, ..., 15]$ in Fig. 6. We show an interesting fact that the best number of iterations is unknown in advance and varies with the datasets. For example, the best $T = 2$ is for Analcatdata while $T = 3$

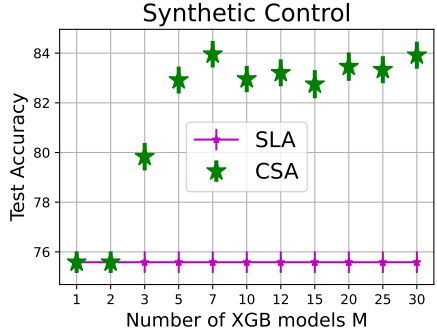 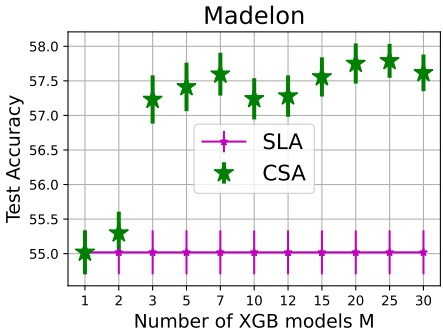

Figure 7: Experiments with different numbers of XGB models for ensembling. The results suggest that using $M \geq 5$ XGBoost models will sufficiently provide a good estimation for the confidence level. Overall, we show that using the confidence score (CSA) will improve the performance than without using it (SLA).

is for Synthetic control, $T = 8$ for Madelon and $T = 2$ for Digits. Our CSA is robust with different choices of $T$ for Synthetic control, Madelon and Digits that our method surpasses the Supervised Learning (SL) by a wide margin. Having said that, CSA with $T > 6$ on Analcatdata can degrade the performance below SL. Therefore, we recommend using $T \leq 5$, and in the main experiment, we have set $T = 5$ for all datasets.

### B.3   Experiments with different number of XGB models $M$ for ensembling

To estimate the confidence score for each data point, we have ensembled $M$ XGBoost models with different sets of hyperparameters. Since we are interested in knowing how many models $M$ should we use for the experiments. We below perform the experiments with varying the number of models $M = [1, ..., 30]$ in Fig. 7. We have shown on two datasets that using $M \geq 5$ will be enough to estimate the confidence score to achieve the best performance.

Additionally, we have demonstrated that making use of the confidence score (the case of CSA) will consistently improve the performance against without using it (the case of SLA). This is because we can ignore and do not assign labels to data points which are in low confidence scores, such as high variance or overlapping between the highest and second-highest class. These uncertain samples can be seen at the dark area in Fig. 2.

### B.4   Computational Time

We analyze the computational time used in our CSA using Madelon dataset. As part of the ensembling task, we need to train $M$ XGBoost models. As shown previously in Appendix B.3, the recommended number of XGBoost models is $M \geq 5$ and we have used $M = 10$ in all experiments. Note that training $M$ XGBoost models can be taken in parallel to speed up the time.

We present in Fig. 8 a relative comparison of the computational times taken by each component in CSA, including: time for training a single model of XGBoost, time for calculating T-test, and time used by Sinkhorn's algorithm. We show that the XGB training takes the most time, T-test estimation and Sinkhorn algorithm will consume less time. Especially, Sinkhorn's algorithm takes an unnoticeable time, such as 0.1 sec for Madelon dataset.

We also observe that the XGBoost computational time used in each iteration will increase with iterations because more data samples are augmented into the labeled set at each pseudo-label iteration. On the other hand, the T-test will take less time with iteration because the number of unlabeled samples reduces over time.

To handle the high complexity due to the growing dataset, one potential solution is to consider online learning to incrementally update the model for XGB, without retraining from the scratch (Montiel et al., 2020).

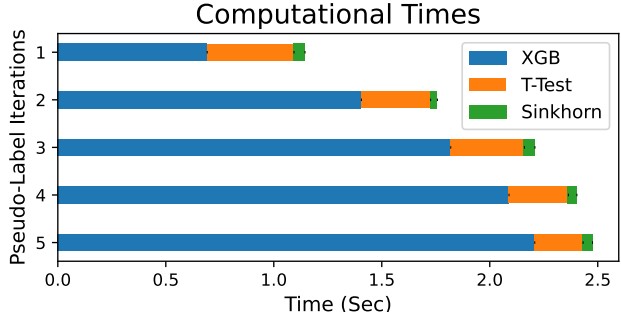

Figure 8: The computational time per each component in CSA for Madelon dataset. We show that the time used for computing both T-test and Sinkhorn is negligible as approximately within a second while the time for XGBoost training takes longer.

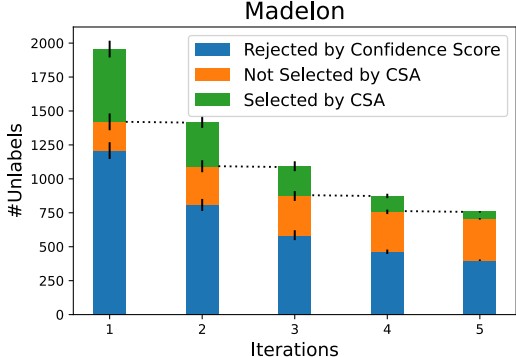

Figure 9: The statistic of the number of samples rejected by the confidence score (t-test), selected by CSA and not selected. The remaining unlabeled data in the next iteration excludes the samples being selected by CSA. Thus, the number of unlabeled samples is reduced across pseudo-label iterations.

Table 6: Comparison on the effect of introducing the lower bound constraint $\mathbf{w}_-$ in the optimal transport formulation. We show that adding the $\mathbf{w}_-$ into the OT formulation will help the performance slightly and adding the confident score will boost the performance significantly.

| Datasets | OT without $\mathbf{w}_-$ SLA (Tai et al., 2021) | OT with $\mathbf{w}_-$ CSA without confidence | OT with $[\mathbf{w}_-, \mathbf{w}^+]$ CSA with confidence |
|---|---|---|---|
| Segment | $95.80 \pm 1$ | $95.96 \pm 1$ | $\mathbf{95.90} \pm 1$ |
| Wdbc | $90.61 \pm 2$ | $91.27 \pm 3$ | $\mathbf{91.83} \pm 3$ |
| Analcatdata | $90.98 \pm 2$ | $91.38 \pm 3$ | $\mathbf{96.60} \pm 2$ |
| German-credit | $70.72 \pm 3$ | $70.93 \pm 3$ | $\mathbf{71.47} \pm 3$ |
| Madelon | $56.53 \pm 4$ | $56.87 \pm 4$ | $\mathbf{57.51} \pm 3$ |
| Dna | $88.09 \pm 2$ | $89.14 \pm 2$ | $\mathbf{89.24} \pm 1$ |
| Agar Lepiota | $58.96 \pm 1$ | $59.12 \pm 1$ | $\mathbf{59.53} \pm 1$ |
| Breast cancer | $92.76 \pm 2$ | $92.81 \pm 2$ | $\mathbf{93.55} \pm 2$ |
| Digits | $81.51 \pm 3$ | $81.68 \pm 3$ | $\mathbf{88.10} \pm 2$ |

Table 7: XGBoost hyperparameters range for ensembling.

| Name | Min | Max |
|---|---|---|
| Learning rate | 0.01 | 0.3 |
| Max depth | 3 | 20 |
| Subsample | 0.5 | 1 |
| Colsample_bytree | 0.4 | 1 |
| Colsample_bylevel | 0.4 | 1 |
| n_estimators | 100 | 1000 |

Table 8: Effect of Ensembling.

| Datasets | PL (single XGB) | PL (ensemble XGBs) | CSA |
|---|---|---|---|
| Wdbc | $91.23 \pm 3$ | $90.22 \pm 3$ | $91.83 \pm 3$ |
| Analcatdata | $90.95 \pm 2$ | $91.84 \pm 3$ | $96.60 \pm 2$ |
| German-credit | $70.72 \pm 3$ | $70.15 \pm 2$ | $71.47 \pm 3$ |
| Breast cancer | $92.89 \pm 2$ | $92.63 \pm 2$ | $93.55 \pm 2$ |
| Digits | $81.67 \pm 3$ | $82.91 \pm 3$ | $88.10 \pm 2$ |

### B.5 STATISTICS ON THE NUMBER OF POINTS REJECTED BY T-TEST AND ACCEPTED BY CSA

We plot the analysis on the number of samples accepted by CSA as well as rejected by the confidence score w.r.t. iterations $t = 1, ..., T$ in Fig. 9. We can see that the time used for training XGBoost model is increasing over iterations because more samples are added into the labeled set which enlarges the training set. Overall, the computational time for XGB and T-test is within a second.

### B.6 EFFECT OF THE LOWER BOUND $\mathbf{w}_-$ FOR THE OPTIMAL TRANSPORT ASSIGNMENT

In the proposed CSA, we have modified the original optimal transport formulation in Tai et al. (2021) to introduce the lower bound $\mathbf{w}_-$ to ensure that the proportion of the assign labels in each class should be above this threshold. This design is useful in practice to prevent from being dominated by the high-frequent class while low-frequent class receives low or zero allocation. We summarize the comparison in Table 6 which shows that introducing the lower bound $\mathbf{w}_-$ will improve marginally the performance (middle column). Then, introducing the confident estimation in CSA will significantly boost the performance (right column).

### B.7 EFFECT OF ENSEMBLING TO THE FINAL PERFORMANCE

Ensembling XGB is part of the main algorithm to estimate the confidence level. In this section, we aim to demonstrate that the final performance improvement comes primarily by other components of CSA, such as confidence score filtering, optimal transport assignment. We run an ablation study to learn the effect of ensembling in isolation. We show in Table 8 that the gain from the ensemble toward the final performance is *not that significant*, as opposed to the uncertainty filtering and OT assignment. In particular, ensembling XGBs will improve 2/5 datasets (Analcatdata, Digits) while degrade slightly for the other 3/5 datasets (Wdbc, German-credit, Breast cancer).

## C APPENDIX: EXPERIMENTAL SETTINGS

We provide additional information about our experiments. We empirically calculate the label frequency vector $\mathbf{w} \in \mathbb{R}^K$ from the training data, then set the upper $\mathbf{w}_+ = 1.1 \times \mathbf{w}$ and lower $\mathbf{w}_- = 0.9 \times \mathbf{w}$ while we note that these values can vary with the datasets. We set the Sinkhorn regularizer $\varepsilon = 0.01$. We use $T = 5$ pseudo-label iterations for the main experiments while we provide analysis with different choices of the iterations in Appendix B.2.

We present the hyperparameters ranging for XGBoost in Table 7. Then, we summarize statistics for all of the datasets used in our experiments in Table 9.

We design to allocate more data points at the earlier iterations and less at later iterations by setting a decreasing vector over time $\rho_t = (T - t + 1)/(T + 1)$, where $T$ is the maximum number of pseudo iterations, which is then normalized $\rho_t = \frac{\rho_t}{\sum_{\forall t} \rho_t}$.

## D APPENDIX: ADDITIONAL EXPERIMENTS

We present a further comparison with the pseudo-labeling baselines in Fig. 10 using 6 additional datasets.

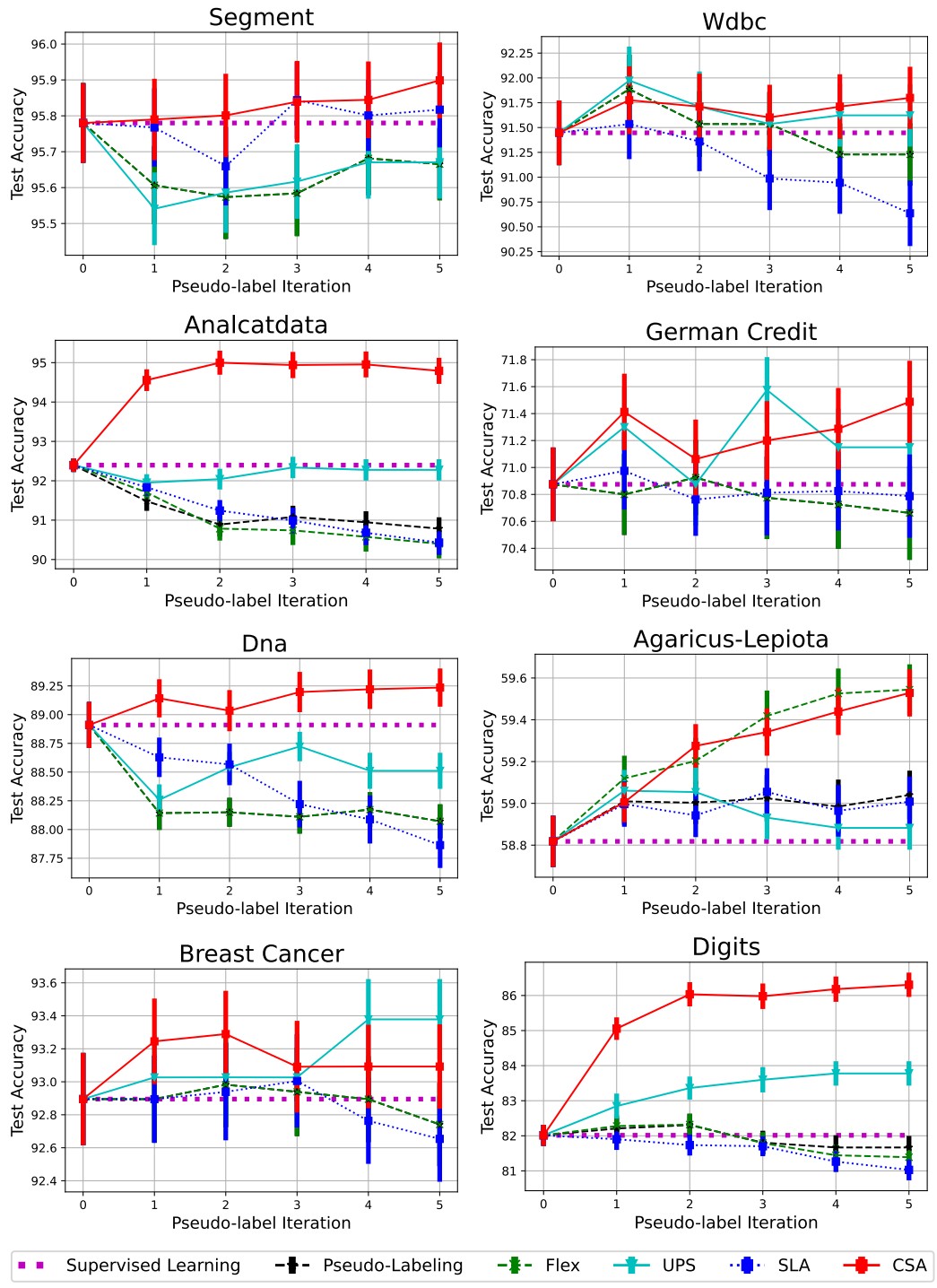

Figure 10: Additional experimental results in comparison with the pseudo-labeling methods on tabular data. The results clearly demonstrate the efficiency of the proposed CSA against the baselines.

Table 9: Dataset statistics including the number of classes $K$, number of feature $d$, number of test samples, number of labeled, unlabeled and the ratio between labeled versus unlabeled (last column).

| Datasets | Domains | #Classes $K$ | #Feat $d$ | #Test | Train | |
|---|---|---|---|---|---|---|
| | | | | | #Labeled | #Unlabeled |
| Segment | image | 7 | 19 | 462 | 739 | 1109 |
| Wdbc | medical | 2 | 30 | 114 | 45 | 410 |
| Analcatdata | economic | 4 | 70 | 169 | 67 | 605 |
| German-credit | finance | 2 | 24 | 200 | 160 | 640 |
| Madelon | artificial | 2 | 500 | 520 | 124 | 1956 |
| Dna | biology | 3 | 180 | 152 | 638 | 2,396 |
| Agaricus lepiota | biology | 7 | 22 | 1625 | 3,249 | 3,249 |
| Breast cancer | image | 2 | 30 | 114 | 91 | 364 |
| Digits | image | 10 | 64 | 360 | 287 | 1150 |
| Cifar-10 | image | 10 | 32 x 32 | 10,000 | 1000 | 49,000 |
| Yeast | biology | 14 | 103 | 726 | 845 | 846 |
| Emotions | audio | 6 | 72 | 178 | 207 | 208 |

