# OpenReview forum: "Confident Sinkhorn Allocation for Pseudo-Labeling"
_ICLR.cc/2024/Conference — Submitted to ICLR 2024_

### Official Review · Reviewer_33FF · 2023-10-24

**Soundness:** 3 good
**Presentation:** 3 good
**Contribution:** 3 good
**Rating:** 8
**Confidence:** 3

**Summary:**

To solve the problem of misallocation caused by the overconfidence of the pseudo-labeling method due to threshold sensitivity,This paper studies theoretically the role of uncertainty to pseudo-labeling and proposes Confident Sinkhorn Allocation (CSA), which identifies the best pseudolabel allocation via optimal transport to only samples with high confidence scores. CSA utilizes Sinkhorn’s algorithm to assign labels to only the data samples with high confidence scores, eliminating the need to predefine the heuristic thresholds used in existing pseudo-labeling methods. In terms of theory,this paper study the pseudo-labelling process when training on labeled set and predicting unlabeled data using a PAC-Bayes generalization bound. CSA specifies the frequency of assigned labels including the lower bound and upper bound per class as well as the fraction of data points to be assigned. Then, the optimal transport will automatically perform row and column scalings. Additionally, this paper proposes to use the Integral Probability Metrics to extend and improve the existing PAC-Bayes bound which relies on the Kullback-Leibler (KL) divergence.

**Strengths:**

This paper explains the label assignment process as an optimal transport problem between examples and classes, and solves it using the confident Sinkhorn algorithm.The proposed CSA is widely applicable to various data domains, and could be used in concert with consistency-based approaches, but is particularly useful for data domain where pretext tasks and data augmentation are not applicable, such as tabular data.

The theoretical result reveals that less uncertainty is more helpful. More number of unlabeled data is useful for a good estimation. Less number of classes and less number of input dimensions will make the estimation easier. The analysis takes a step further to show that both aleatoric uncertainty and epistemic uncertainty can reduce the probability of obtaining a good estimation.

In the experiment, all models use the same backbone network, and the settings of the models are described in comparison to it. Experimental verification shows that optimal transport cannot be achieved simply by changing the threshold value.The paper also conducts other empirical analysis which can be sensitive to the performance.

**Weaknesses:**

The legends of the figures in the paper are not very clear, making it difficult to understand the meaning of the various elements in the figures without reading the relevant paragraphs in detail.

The part about CSA in the article and the part about PAC-Bayes bound seem to have insufficient connection. Even without using knowledge about the PAC-Bayes bound, the description and derivation of the CSA part still seem to hold.

**Questions:**

Refer to Weaknesses.

---

> ### Author Response · Authors · 2023-11-15
> **author response**
>
> Thank you for your positive review and precisely highlighting all of our key contributions. We hope our responses to the minor concerns provide the necessary reassurance required.
>
> ---
>
> Question: Connection between CSA and PAC-Bayes bound in the algorithm.
>
> Answer: the key label assignment in pseudo-labeling (PL) is to train on labeled data and generalize this estimation to predict the unlabeled data. Most of the existing research in PL implicitly assumes the strong generalization that the model performance on the labeled (train set) and unlabeled (test set) are similar. However, this generalization assumption is not always true. For example, if we have limited samples in the labeled set, the model can not generalize to predict the unlabeled set well. Thus, it can lead to poor PL performance.
> Without the PAC-Bayes result, the CSA part may fail into the above generalization issue.
>
> Therefore, we present the PAC-Bayes result to theoretically analyze the generalization performance of ensemble models trained on labeled data and tested on unlabeled data. In particular, Theorem 2 reveals that the more number of labels data $N_l$ makes the generalization error tighter (better). This generalization finding is consistently aligned with the result in Theorem 1 (analyzing from a different perspective). In addition, our generalization result in Theorem 2 considers the ensemble of multiple classifiers – the setting which has not been analyzed in Theorem 1. Thus, both theoretical results complement and strengthen each other in our paper.
>
> ---
>
> Minor suggestion for legend:
> Thank you for mentioning this. We will improve the presentation of the legends in the final version.Particularly, we will make better word choices for the legend and have the corresponding caption to be more descriptive.

---

> ### Comment · Reviewer_33FF · 2023-11-23
> **Response**
>
> Thanks for the reponse. I have no further concerns to this work, so I prefer to maintain my score, while I'm not very familiar with the uncertainty estimation methonds, such that I suggest that Chairs mainly refer to the opinions of reviewers who are more professional in this field.

---

### Official Review · Reviewer_jWfa · 2023-10-30

**Soundness:** 3 good
**Presentation:** 3 good
**Contribution:** 3 good
**Rating:** 6
**Confidence:** 3

**Summary:**

Pseudo-labeling is suitable for learning without any domain assumptions. This work proposes a Pseudo-Labeling algorithm based on a confident score derived from ensemble models.  Firstly, a mathematical proof is presented to elucidate the impact of uncertainty in classifiers. Welch’s T-test is employed to ascertain whether the most likely class holds statistically greater significance than the second-most likely class. This serves to diminish uncertainty in the estimated classifiers. Subsequently, the label allocation process is transferred to the optimization of the optimal transport problem, and the Sinkhorn Algorithm is employed to swiftly approximate the solution. Moreover, this study establishes PAC-Bayes results using Integral Probability Metrics, which provides a guarantee of generalization performance. Additionally, comprehensive experiments are devised to facilitate a comparative evaluation with other relevant works in the field.

**Strengths:**

1. This work introduces an efficient algorithm aimed at mitigating uncertainty in pseudo-labeling. It leverages ensemble models to assess the confidence of labeling. Additionally, a comprehensive experimental setup is designed, encompassing not only accuracy comparisons with state-of-the-art algorithms but also evaluations across various dimensions.
2. This work provides a solid mathematical proof for uncertainty analysis in Pseudo-Labeling and extends PAC-Bayes bounds to ensemble models, both of which contribute to subsequent research in this domain.

**Weaknesses:**

1. Errors are present in the tables and figures. In Table 1, it is noted that in the comparison of related approaches, FlexMatch should be characterized as non-greedy based on the provided content. Regarding Figure 4, the top red square on the left fails to adequately illustrate the distinctions in assignments.
2. In the section pertaining to the analysis of uncertainty in Pseudo-Labeling (PL), some aspects of the formulation concerning the settings are found to be incomplete. Consequently, this has resulted in certain points of confusion in comprehension. Although the appendix contains proofs that address some of my queries, further elucidation may be beneficial.
3. In Algorithm 1, Confident Sinkhorn Label Allocation (simplified), the derivation of b_{-} and b_{+} when setting marginal distributions is not explicitly stated. Based on the content, it is inferred that they are empirically estimated from the class label frequency in the training data or from prior knowledge. This should be explicitly mentioned in the algorithm; otherwise, it leads to unknowns and incompleteness in the algorithm.

**Questions:**

1. In Section 2.2, you outlined two challenges associated with assigning pseudo-labels. The proposed resolution entails employing an ensemble learning framework along with Welch's T-test to discern and exclude less confident samples. Are there alternative, more dependable methods for comparing the most probable class with the second most probable class? (Except those compared in the appendix)
2. In the appendix, it is noted that the computational time of XGBoost increases with each iteration. Could this escalation in computational time become substantial when applied on a larger scale, potentially resulting in inefficiencies that outweigh its benefits?
3. I've noted that in the algorithm when ρ equals 1, it indicates full allocation. Additionally, I observed a limited elucidation regarding this parameter. In the appendix, ρ is configured to allocate more data points in the earlier iterations and fewer in the later ones. I am intrigued by the potential impact of varying ρ. While this obviates the necessity to predefine a suitable threshold γ, it introduces a new variable, ρ, necessitating a predefined value. Is the outcome sensitive to the choice of ρ? Is there a universally applicable ρ that ensures the algorithm's effectiveness across diverse tasks? The role of ρ in the algorithm appears somewhat ambiguous, and I am particularly keen on gaining a deeper understanding of it.

---

> ### Author Response · Authors · 2023-11-15
> **thank you**
>
> We thank the reviewer for insightful suggestions which are very helpful to improve the clarity of the paper. When reading this, it seems the reviewer believes our work makes a meaningful contribution. We hope we can clarify your issues.
>
> ---
> Question: In Table 1, FlexMatch should be characterized as non-greedy
>
> Answer: Using OT, each datapoint will be jointly assigned labels in the presence of other assignments. In contrast, using FlexMatch (and other PL approaches), each data point assignment will purely depend on the given threshold and does not necessarily depend on the other assignments. FlexMatch still falls into this greedy category.
> Nevertheless, in comparison to the other PL approaches, FlexMatch is less greedy in the sense that it defines the different thresholds for each class.
>
> ---
>
> Question: Figure 4, the top red square on the left fails to adequately illustrate the distinctions in assignments.
>
> Answer: We can improve this figure in the final version to make it clearer. In Figure 4, we use color (yellow is 1 and dark blue is 0) to indicate the numerical value of the assignment. The top red square actually has the value of 0.3 in brighter blue, although it is not visually clear enough to distinguish the dark blue (value =0), comparing to the other red squares in the bottom.
>
> ---
>
> Question: In Algorithm 1, CSA (simplified), the derivation of $b_{-}$ and $b_{+}$ ...it is inferred that they are empirically estimated from the class label frequency in the training data or from prior knowledge.
>
> Answer: Yes, they are empirically estimated from the class label frequency in the training data or from prior knowledge. We will mention them in the algorithm for clarity. Note that, $b_{-}$ and $b_{+}$ should be correctly written as $w_{-}$ and $w_{+}$.
>
> ---
> Question: Are there alternative methods for statistical testing, more dependable methods for comparing the most probable class with the second most probable class?
>
> Answer: Other alternatives for statistical testing include: Student-t test and Mann–Whitney U test.
> However, according to the literature [1,2,3], we should use Welch’s t-test by default, instead of Student’s t-test, because Welch's t-test performs better whenever sample sizes and variances are unequal between groups, and gives the same result when sample sizes and variances are equal.
> In addition, Ruxton (2006) [3] stated “...the unequal variance t-test (or Welch’s t-test should always be used in preference to the Student’s t-test or Mann–Whitney U test”
>
> [1] Delacre, M. et al (2017). Why Psychologists Should by Default Use Welch’s t-test Instead of Student’s t-test. International Review of Social Psychology
>
> [2] https://daniellakens.blogspot.com/2015/01/always-use-welchs-t-test-instead-of.html
>
> [3] Ruxton, G. D. (2006). The unequal variance t-test is an underused alternative to Student's t-test and the Mann–Whitney U test. Behavioral Ecology.
>
> We, therefore, opt for the proposed Welch’s t-test as the default choice in our framework. However, we can bring the above discussion in the paper to make it clearer for the readers. We thank the Reviewer for raising this useful question.
>
> ---
> Question: the computational time of XGBoost increases with each iteration
>
> Answer: First of all, most of the pseudo-labeling and SSL-based methods will suffer the same issue that the labeled dataset will grow over time after augmenting the unlabeled data. Such a growing dataset size will lead to more computational time.
> One potential solution is to consider online learning to incrementally update the model for XGB, without retraining from the scratch [1]. Note that this is beyond the scope of our paper, but we can mention it in the updated version of the paper.
>
> [1] Montiel, et al. "Adaptive xgboost for evolving data streams. IJCNN 2020
>
> ---
> Question: $\rho$ obviates the necessity to predefine a suitable threshold $\gamma$. But it introduces a new variable, $\rho$, necessitating a predefined value.
>
> Answer: This is a very insightful comment. We follow the existing setting from SLA [44] to define the allocation schedule $\rho$ controlling the fraction of examples to be assigned labels in each iteration. The role of $\rho$ is to avoid assigning the entire 100% allocation in the first iteration for the OT approaches (both CSA and SLA). On the other hand, the threshold will stop this behavior from happening for other PL approaches.
>
> A single assignment in PL can be sensitive to the value of $\gamma$ in being assigned or not. On the contrary with $\rho$ in OT, the assignment will be less sensitive to $\rho$ because we optimize the objective function to learn the assignment, i.e., minimizing the cost matrix to satisfy the row marginal distribution and column marginal distribution given the schedule allocation. There exist other factors (defined in the objective function) that influence the optimal assignment than the value of $\rho$ itself.

---

### Official Review · Reviewer_EorJ · 2023-10-31

**Soundness:** 2 fair
**Presentation:** 2 fair
**Contribution:** 2 fair
**Rating:** 5
**Confidence:** 4

**Summary:**

This paper introduces Confident Sinkhorn Allocation (CSA) as an approach to improve pseudo labeling (PL) in the context of semi-supervised learning. The author delves into an analysis of the uncertainties associated with pseudo-labeling and introduces optimal transport as a means to mitigate the sensitivity observed in Greedy PL. Additionally, the paper presents a PAC-Bayes generalization bound that incorporates Integral Probability Metrics.

**Strengths:**

The issue of excessive confidence and sensitivity to thresholds in pseudo labeling (PL) is indeed intriguing. The author conducts an analysis of the uncertainties within PL and offers some insights into this matter.

**Weaknesses:**

1.	The paper's contribution appears somewhat vague. While it introduces a new pseudo labeling (PL) method, it lacks a clear probabilistic formulation. However, it's worth noting that the author does provide a PAC-Bayes generalization bound in Section 2.4, particularly for ensembling multiple classifiers. It would enhance clarity to explicitly state the individual contributions of various sections within the methodology.

2.	There seems to be an inconsistency in the citation format used in the main text. The citation style in the introduction relies on numbers, but corresponding numbers in the reference list are missing. This inconsistency makes it challenging to match citations in the main text to the references.

3.	The novelty of the optimal transport assignment in Section 2.3 appears somewhat limited. The primary concept seems to be derived from the original SLA [44]. Clarifying the extent of novelty and how the proposed method builds upon or diverges from previous work would be beneficial for the reader's understanding.

**Questions:**

The method is primarily evaluated within the context of semi-supervised learning tasks, where sample selection is a general aspect of the approach. It raises the question of whether this technique can be extended to other settings, such as active learning or learning with noisy labels. Exploring the adaptability of this method to these different scenarios and discussing potential challenges or advantages would provide valuable insights into its broader applicability and limitations.

---

> ### Author Response · Authors · 2023-11-15
> **author response**
>
> Thank you for the detailed review and a wide range of interesting comments. This review highlighted areas of our paper that absolutely needed to be clarified, so we appreciate the considerable time spent in finding them. We thank the reviewer in advance for reading our response, and hope that we have provided enough clarity that it is possible to provide us with a higher score as a consequence.
>
> ---
>
> Question: It would enhance clarity to explicitly state the individual contributions of various sections within the methodology. The novelty of the optimal transport assignment in Section 2.3 appears somewhat limited. The primary concept seems to be derived from the original SLA [44].
>
> Answer: The OT assignment plays an important role in our algorithm. However, we don't claim the novelty and contribution toward this OT part (we have clearly stated in the abstract, the last paragraph of the introduction, and Sec 2.3).
>
> We summarize the novelty and significance as follows:
>
> First, we are the first (to the best of our knowledge) to theoretically study the role of uncertainty in PL.
> The theoretical finding reveals that less uncertainty is more helpful. More unlabeled data is useful for a good estimation. Less number of classes and less number of input dimensions will make the estimation easier. The analysis takes a step further to show that both aleatoric uncertainty and epistemic uncertainty can reduce the probability of obtaining a good estimation.
> We note that the result presented in SLA [44] has only analyzed the first property (more number of unlabeled data is useful for a good estimation). The rest of the theoretical finding is novel and original.
>
> Second, we propose to use Welch’s T-test to ascertain whether the most likely class holds statistically greater significance than the second-most likely class. This T-test (to be used with OT)  eliminates the need to predefine the heuristic thresholds used in existing pseudo-labeling methods.
> Third, we extend PAC-Bayes bounds of ensemble models which provides a guarantee of generalization performance when trained on labeled data and tested on unlabeled data. Our bound strictly improves existing PAC-Bayes bounds for ensemble models and is of extended independent interest.
>
> Note that  both of our theoretical results contribute to subsequent research in this domain. These are identified and  highlighted by Reviewer jWfa  and Reviewer 33FF.
>
> ---
>
> Question: There seems to be an inconsistency in the citation format used in the main text. The citation style in the introduction relies on numbers, but corresponding numbers in the reference list are missing. This inconsistency makes it challenging to match citations in the main text to the references.
>
> Answer: We thank the reviewer for pointing out this citation format. In the current version, when clicking to the number citation in the main text using PDF, it will automatically link to the correct citation in the Reference. We will fix this formatting and update the paper.
>
> ---
>
> Questions: The method is primarily evaluated within the context of semi-supervised learning tasks, where sample selection is a general aspect of the approach. It raises the question of whether this technique can be extended to other settings, such as active learning or learning with noisy labels. Exploring the adaptability of this method to these different scenarios and discussing potential challenges or advantages would provide valuable insights into its broader applicability and limitations.
>
> Answer: we would like to thank the reviewer for the suggestion on the possible research extension from our proposed method, beyond semi-supervised learning tasks. Indeed, active learning or learning with noisy labels are the promising research directions that can leverage pseudo-labeling. Within the scope of our paper, we focus on the main task of SSL in which most of the pseudo-labeling methods have been evaluated. We will consider active learning and learning with noisy labels as the future work.

---

> > ### Comment · Reviewer_EorJ · 2023-11-16
> >
> > Thanks for the detailed clarification of the contribution in this work. Please update the manuscript in Author Console.

---

> > > ### Author Response · Authors · 2023-11-17
> > > **Manuscript updated**
> > >
> > > Hi Reviewer EorJ, thank you for the response. We have updated the manuscript as requested.
> > >
> > > We believe we have addressed most of the concerns raised. As such, we are optimistic that you would be able to provide us with a higher score. If this is not the case, please let us know so we have an opportunity to address the remaining concerns. Thank you very much.

---

> > > > ### Comment · Reviewer_EorJ · 2023-11-22
> > > >
> > > > Thanks. I raised the score to 5.

---

### Official Review · Reviewer_Pwpb · 2023-10-31

**Soundness:** 2 fair
**Presentation:** 2 fair
**Contribution:** 2 fair
**Rating:** 5
**Confidence:** 4

**Summary:**

The paper explores the theoretical aspects of incorporating uncertainty in pseudo-labeling and introduces a new method, Confident Sinkhorn Allocation (CSA). CSA aims to determine the best pseudo-labels using optimal transport, focusing on samples with high confidence levels. Additionally, the study suggests utilizing Integral Probability Metrics to enhance and refine the current PAC-Bayes bound, which is dependent on the Kullback-Leibler (KL) divergence, for ensemble models. Experimental results indicate CSA's superiority over existing methods in semi-supervised learning.

**Strengths:**

1. The authors incorporate uncertainty into the pseudo-labeling generation process and provide a theoretical interpretation.
2. The authors study theoretically the pseudo-labelling process when training on labeled set and predicting unlabeled data using a PAC-Bayes generalization bound.

**Weaknesses:**

1. The choice to employ optimal transport methods for pseudo-labeling is not immediately clear, especially given the existence of the method detailed in section 2.2. It would be beneficial if the authors could elucidate on the rationale behind selecting optimal transport over the direct pseudo-labeling approach from section 2.2.
2. The paper employs an ensemble of M models, but it is ambiguous whether the observed improvement in performance is attributed to the ensemble effect or the proposed algorithm itself. The absence of an ablation study leaves this point unclarified.
3. The optimization objectives for optimal transport presented in section 2.3 lack clear explanations. It would be beneficial if the authors provide detailed interpretations for each constraint, elucidating on their roles and significance within the context of the problem.
4. In section 2.1 of the article, there appears to be some inconsistency and potential oversight regarding notation. Firstly, the representation of the unlabeled data set as $\{\tilde{X_i^k}\}$ seems non-standard. Given that $X_i^k$ denotes an individual data point, it would be more appropriate to use lowercase notation for clarity. Secondly, the expression for the probabilistic classifiers, specifically $f_k(x_i)$, is stated to produce a scalar value indicating the likelihood of $x_i$ being labeled as $k$. However, the provided formulation $f_k(x_i):=\mathcal{N}(x_i|\hat\theta_k,\Lambda)$ suggests it's a function mapping to a normal distribution parameterized by $x_i\vert\hat\theta_k$ and $\Lambda$.
5. In Theorem 1, the definition of $\mu_{\backslash k}=\mu_j\vert\exist j\in {1,...K}\backslash k$ is ambiguous. It would be beneficial for the authors to provide a more explicit definition or clarification regarding the intended meaning of $\mu_{\backslash k}$ in the context of the theorem.

**Questions:**

1. Why did the authors choose to use optimal transport methods for pseudo-labeling instead of the direct approach described in section 2.2?
2. How is the pseudo-labeling method described in section 2.2 related to the optimal transport pseudo-labeling approach in section 2.3? How do these two methods interact or complement each other in the overall framework?
3. Have the authors considered conducting an ablation study to discern the individual contributions of the ensemble effect and the proposed algorithm to the overall performance improvement?

---

> ### Author Response · Authors · 2023-11-15
> **author response**
>
> We thank the Reviewer for the overall positive comments and mentioning a few points to be clarified. We are glad that the Reviewer recognized our theoretical contribution for characterizing the role of uncertainty in PL and generalization bound.
>
> ---
>
> Q: The optimization objectives for OT in Sec 2.3 lack clear explanations
>
> A: Given the space constraints, we have presented additional details of the OT derivation and optimization in Appendix A.2 and A.3.
> There are three key components in the OT setting: the cost matrix, the marginal distributions for row and for column. We minimize the cost matrix to learn the optimal assignment, satisfying both marginal distributions. Specifically, the column marginal is a vector of one, i.e., each data point should not be assigned to more than one label, Eq. (6) and Eq. (11). In addition, each label should receive the amount of assignment ranging from lower bound and upper bound label frequency $[w_-, w_+]$, Eq. (7,8,12,13). We will highlight this interpretation in the final version.
>
> ---
>
> Q: In Sec 2.1. the representation of the unlabeled data set as $\tilde{ X^k_i }$ seems non-standard. Given that $X^k_i$ denotes an individual data point, it would be more appropriate to use lowercase notation for clarity.
>
> A: we follow the same notation in [Yang and Xu (2020), Sec 2.1] to denote the labeled vs unlabeled data set. The difference is that the original paper considers the binary setting with $X^+$ and $X^-$ while our setting considers multi-classification with k classes. In addition, we (also by Yang and Xu) use $X^k_i$ to define a data point which takes a class k. $X^k_i$ is equivalent to write $x_i \in X^k$ and $\tilde{X}^k_i$ is $x_i \in \tilde{X^k}$.
>
> ---
>
> Q: the expression for the probabilistic classifiers $f_k(x_i)$ stated to produce a scalar value indicating the LLK of $x_i$ being labeled as $k$. However, the provided formulation $f_k(x_i) = N( x_i | \hat{ \theta_k} , \Lambda)$ suggests it's a function mapping to a normal distribution parameterized by $x_i |  \hat{ \theta_k}$ and $\Lambda$.
>
> A: $f_k(x_i)$ produces a scalar value indicating the LLK of $x_i$ being labeled as $k$. Then, we particularly consider the simple case of the Gaussian classifier, $f_k(x_i) := p(x_i | y_i = k) = N( x_i | \hat{ \theta_k} , \Lambda)$. This is the common notation used in [1].
>
> [1] https://www.cs.ubc.ca/~murphyk/Teaching/CS340-Fall07/gaussClassif.pdf (see slide 7)
>
> ---
>
> Q: In Theorem 1, $\mu_{\\k} = \mu_j | \exists j \in {1…K} \ k$ is ambiguous
>
> A: Intuitively,  k \ {1,...K}  indicates the correct class index. We denote the incorrect class being \k = \mu_j \in {1…K} \ k. This refers to one of the remaining classes excluding k. $\exists j$ can be optionally dropped to make it concise. We hope it clear and will clarify them in the context of the theorem
>
> ---
>
> Q: ablation study to discern the individual contributions of the ensemble effect and the proposed algorithm to the overall performance improvement?
>
> A: Thanks for the great suggestion. We have made (below) the ablation study to learn the effect of ensembling. We show that the gain from the ensemble toward the final performance is not that significant to the overall performance improvement. In particular, ensembling XGBs will improve 2/5 datasets (Analcatdata, Digits) while degrade slightly for the other 3/5 datasets.
>
>
> | Dataset | PL (single XGB) | PL (ensemble XGBs) | CSA |
> | ------ | ------| ------| ------|
> | Wdbc | 91.23±3 | 90.22±3 | 91.83±3 |
> | Analcatdata  | 90.95±2 | 91.84±3 | 96.60±2 |
> | German-credit | 70.72±3 | 70.15±2 | 71.47±3 |
> | Breast cancer | 92.89±2 | 92.63±2 | 93.55±2 |
> | Digits | 81.67±3 | 82.91±3 | 88.10±2 |
>
>
> ---
>
> Q: the rationale behind selecting OT over the direct PL approach (Sec 2.2)
>
> A: The Reviewer mentions pseudo-labeling approach presented from Sec 2.2. However, Sec 2.2. presents Welch's T-test for identifying high confidence samples while the PL approaches are described in the Related Work subsection of the Introduction. Please correct us if we miss anything.
>
> OT is the non-greedy algorithm which can learn the best (non-trivial) label assignments. On the other hand, most of the existing PL methods are greedy which will rely on the predefined threshold to assign the labels. Specifically, using OT, each data point will be jointly assigned labels in the presence of others. In contrast, using the PL, each data point assignment will purely depend on the given threshold and does not necessarily depend on the other assignments.
>
> We show in Sec 3.3 and Fig. 4 that OT assignment cannot be achieved simply by changing the threshold value (also mentioned by the Reviewer 33FF).
>
> ---
>
> We feel we have addressed all of the concerns raised. As such, we are optimistic that you would be able to raise your score. If this is not possible, please let us know so we have an opportunity to address the remaining concerns.

---

> ### Comment · Reviewer_Pwpb · 2023-11-22
>
> Thank you for your response, which has resolved most of my concerns. However, regarding Weakness 4, you mentioned that $ f_k(x_i) := p(x_i | y_i=k) $, but shouldn't it be defined as $ f_k(x_i) := p(y_i=k | x_i) $? f_k produces a scalar value indicating the likelihood of being labeled as k. This definition somewhat affects the credibility of the conclusions drawn in the article, so I have decided to maintain the original score.

---

> ### Author Response · Authors · 2023-11-22
> **Follow up**
>
> We are glad that we have resolved most of the concerns as acknowledged by the reviewer.
>
> We thank the reviewer for mentioning the remaining concern left.
>
> In our paper, we define $f_k$ produces a scalar value indicating the likelihood of being labelled as $k$. In particular, in Line 2, Page 4 in the paper, we have defined $f_k(x_i) := N( x_i | \hat{\theta}_k, \Lambda ) $  which is correct.
>
> However, during our response, we have written $f_k(x_i) := \textcolor{red}{p(x_i | y_i = k)} = N( x_i | \hat{\theta}_k, \Lambda ) $ which is confusing as the reviewer pointed out. This should be corrected (by the reviewer) as $f_k(x_i) := \textcolor{blue}{p(y_i = k | x_i)} = N( x_i | \hat{\theta}_k, \Lambda ) $.
>
> Nevertheless, we want to call out that this typo $ \textcolor{red}{p(x_i | y_i = k)}$ is from our rebuttal response while the equation presented in paper is still correct (Line 2, Page 4).
>
> Given (1) the above correctness and (2) that we have addressed all other concerns, we would greatly appreciate if you can reconsider the score or please let us know what is your suggestion to fix this typo?
>
> Thank you!

---

### Author Response · Authors · 2023-11-15
**general response**

Dear,

We would like to thank all of the reviewers for their time and efforts providing comments about the paper. The reviewers have noticed our main objective, which is to propose an “_efficient algorithm aimed at mitigating uncertainty in pseudo-labeling_” (**Reviewer jWfa**) that utilises “_uncertainty into the pseudo-labeling generation process and provides a theoretical interpretation_” (**Reviewer Pwpb**) followed by a “_comprehensive experimental setup_” (**Reviewer jWfa**) that is “_applicable to various data domains, and could be used in concert with consistency-based approaches_” (**Reviewer 33FF**). In addition, we provided a “_solid mathematical proof for uncertainty analysis in Pseudo-Labeling and extends PAC-Bayes bounds to ensemble models_” (**Reviewer jWfa**).

We also acknowledge several clarifying points raised by the Reviewers, e.g., the advantage of OT against the existing PL assignment (**Reviewer Pwpb**) and the novelty and contribution (**Reviewer EorJ**), for which we have addressed in the individual responses.

Best regards,

The authors

---

> ### Author Response · Authors · 2023-11-16
> **Updated the PDF**
>
> As per the Reviewer EorJ suggestion, we have updated the paper reflecting the changes to address the review feedbacks.
>
> In the revised PDF, we use the **orange color** to indicate the new content (page 4, 10, 15, 16, 21,23). We aware that it is slightly over 9 pages, but we will fit the page limit later.
>
> Please let us know so we have an opportunity to address the remaining concerns. Thank you very much!

---

### Author Response · Authors · 2023-11-22
**Message to Reviewers**

Dear Reviewers,

we made the changes you had asked for and uploaded a revision of our paper on the 16th of November.

We were wondering if you had the chance to review our changes and if you are happy with it. It would be great to get some feedback as to whether the changes we made were all done appropriately.

We thank you for taking the time to review our paper and greatly appreciate your feedback.

Best regards,
The Authors

---

### Meta-Review · Area_Chair_yWhe · 2023-12-04

**Metareview:**

This paper worked on pseudo-labeling for semi-supervised learning. It studied the role of uncertainty to pseudo-labeling and proposed Confident Sinkhorn Allocation, which identifies the best pseudo-label allocation via optimal transport to only samples with high confidence scores. The paper received diverged ratings from 5 to 8 (initially from 3 to 8). During the discussion, the most positive reviewer (33FF) emphasized that
> My concerns about this work are solved by the author, but as I mentioned in the response to the authors, I'm not very familiar with the uncertainty estimation methods, such that I suggest that Chairs mainly refer to the opinions of reviewers who are more professional in this field.

Therefore, I think it may be more appropriate to down-weight the opinion of this reviewer and reject this submission for publication.

I personally think that the submission is promising with good enough novelty and significance. The only reason for rejection is its unclear motivation. As mentioned in the abstract,
> PL is sensitive to a threshold and can perform poorly if wrong assignments are made due to overconfidence.

However, since the proposal of FixMatch, there are many papers working on using a dynamic or adaptive threshold to improve upon a fixed threshold, such as "Dash: Semi-Supervised Learning with Dynamic Thresholding" (ICML 2021), and then "Class-Imbalanced Semi-Supervised Learning with Adaptive Thresholding" (ICML 2022) where the dynamic or adaptive thresholds are designed to be class-dependent instead of a single global dynamic or adaptive threshold. The key question is thus what is missing or what is wrong in those dynamic or adaptive thresholding methods.

The answer to the above question is the real motivation of the current submission. Given that those methods were not cited or conceptually/experimentally compared with, I do think this promising submission is not ready in its current version and needs a major revision.

**Justification For Why Not Higher Score:**

The only reason for rejection is its unclear motivation.

**Justification For Why Not Lower Score:**

N/A

---

### Decision · Program_Chairs · 2024-01-16

Reject